# Alethia: A Foundational Encoder for Voice Deepfakes

**Yi Zhu** [1]   **Brahmi Dwivedi** [1]   **Jayaram Raghuram** [1]   **Surya Koppisetti** [1]

## Abstract

Existing voice deepfake detection and localization models rely heavily on representations extracted from speech foundation models (SFMs). However, downstream finetuning has now reached a state of diminishing returns. In this paper, we shift the focus to pretraining and propose a novel recipe that combines *bottleneck masked embedding prediction with flow-matching based spectrogram reconstruction*. The outcome, *Alethia*, is the first foundational audio encoder for various voice deepfake detection and localization tasks. We evaluate on 5 different tasks with 56 benchmark datasets, and note *Alethia* significantly outperforms state-of-the-art SFMs with superior robustness to real-world perturbations and zero-shot generalization to unseen domains (e.g., singing deepfakes). We also demonstrate the limitation of discrete targets in masked token prediction, and show the importance of *continuous embedding* prediction and *generative pretraining* for capturing deepfake artifacts.

## 1. Introduction

State-of-the-art speech foundation models (SFMs), such as Wav2vec (Baevski et al., 2020), WavLM (Chen et al., 2022), and HuBERT (Hsu et al., 2021), provide efficient audio representations that facilitate strong downstream performance on a variety of learning tasks (Yang et al., 2021). At the core of these models is the self-supervised learning paradigm built on a well-designed pretraining recipe and a massive corpus of unlabeled data. A standard pretraining recipe is the BERT-like masked token prediction, where *discrete targets* computed from the speech sample provide a supervisory signal for the representation learning (Baevski et al., 2020; Chen et al., 2022; Hsu et al., 2021; Huzaifah et al., 2024; Li et al., 2023; Yang et al., 2025). This was found to be effective for a variety of audio encoders, for example, the MERT model for music (Li et al., 2023), SPEAR for speech

and audio (Yang et al., 2025), and MERaLiON for Singlish data (Huzaifah et al., 2024), among others.

Of particular interest in this paper is the representation learning for voice deepfakes, which have now become a major concern due to the rapid emergence of high-quality generative methods. Related learning tasks include speech deepfake detection (Li et al., 2025a), singing voice deepfake detection (Zang et al., 2024), partially fake speech localization (He et al., 2025), and deepfake source tracing (Klein et al., 2024). While the downstream finetuning architectures vary across different state-of-the-art (SOTA) models, SFMs remain as the key integral component (often termed as 'frontend'), wherein a better encoder representation yields a strong boost in performance (Yu et al., 2021; Zhang, 2022; Zhang et al., 2025).

Most voice deepfake models today reuse off-the-shelf SFMs, and focus primarily on the downstream finetuning task for performance gains (Zhang et al., 2025). However, recent recent trends suggest that despite finetuning on massive volumes of real and fake speech, models still lack generalizability to unseen generation methods (Ge et al., 2025; Chandra et al., 2025) and are sensitive to real-world perturbations (Li et al., 2025b; Zhu et al., 2025; Müller et al., 2025). These trends suggest that the finetuning paradigm may be fundamentally limited by the representations inherited from the general-purpose SFMs, which are not optimized for capturing generative artifacts. A few studies have recently explored altering the pretraining recipe instead, such as continual pretraining with vocoded speech (Wang & Yamagishi, 2024) and low-rank adaptation of the encoder (Hao et al., 2025), but the reported performance gains were only incremental (Wang & Yamagishi, 2024; Ge et al., 2025). We note that the pretraining objective formulation as well as the scale and variety of pretraining data need to be revisited in order to fully capture the discriminative artifacts in voice deepfakes and thereby achieve better generalization.

In this work, our goal is to build *a foundational encoder for voice deepfakes* that well encapsulates the discriminative patterns between real and generated voice, demonstrates robustness to real-world perturbations and, with simple finetuning, generalizes well to unseen synthesis methods. We propose a pretraining recipe, wherein, the encoder receives supervision from two branches in parallel. In the first branch,

---

[1]Reality Defender. Correspondence to: Yi Zhu <yi.zhu@inrs.ca>.

*Proceedings of the 43rd International Conference on Machine Learning*, Seoul, South Korea. PMLR 306, 2026. Copyright 2026 by the author(s).

the encoder is tasked to perform *bottleneck masked embedding prediction*, where the target embeddings are extracted by a frozen teacher model from unmasked waveforms, and the student model learns to predict different teacher layers from its bottleneck (i.e., layer-averaged) representation obtained from masked waveforms. In the second branch, we introduce a *flow-matching decoder* that generates the unmasked spectrogram conditioned on the bottleneck representation. Together, these objectives formulate a strong pretraining recipe for a foundational encoder for voice deepfake tasks.

Our main contributions are summarized as follows:

**Contribution 1:** We introduce *Alethia*, the first foundational encoder for voice deepfakes. Alethia markedly outperforms commonly used SFMs on 56 datasets across 5 different voice deepfake tasks.

**Contribution 2:** We show the limitation of existing pretraining methods, and propose a new pretraining strategy, consisting of bottleneck masked embedding prediction and flow-matching based spectrogram generation to jointly enhance generalization and robustness to unseen attacks.

**Contribution 3:** We show that Alethia demonstrates zero-shot generalizability to completely unseen attacks (e.g., singing voice deepfakes) and is robust to various real-world perturbations without needing complex augmentations during finetuning.

## 2. Related Works

### 2.1. Predictive Pretraining and Target Choice

Mainstream SFMs usually rely on predictive pretraining, where the predictive targets are pseudo-labels/features extracted from the signal itself (Mohamed et al., 2022). Among different pretraining components (e.g., target, model architecture, data, etc.), the target choice has been demonstrated to be critical for model generalizability (Mohamed et al., 2022; Assran et al., 2023; Balestriero & LeCun, 2024). Predominant encoders in the deepfake domain, such as Wav2vec2 (Baevski et al., 2020), WavLM (Chen et al., 2022), and HuBERT (Hsu et al., 2021), apply quantization via codebook or clustering techniques to generate discrete tokens as targets. While discretization effectively extracts high-level phonetic structures and filter out attributes less relevant to speech content or speaker (Baevski et al., 2020; Hsu et al., 2021; Mohamed et al., 2022), other signal nuances such as generation traces of voice deepfakes could also be overlooked. More recently, masked embedding prediction has shown promising results across audio and vision domain. Some representative works include Data2vec2 (Baevski et al., 2023), JEPA (Assran et al., 2023), and V-JEPA (Assran et al., 2025), which leverage self-distillation to align

student and teacher output in the latent space. While in voice deepfake tasks, masked token prediction models still yield leading performance (Li et al., 2025a), we hypothesize that reconstruction in the continuous embedding space could provide a more granular objective for modeling generation traces and have the potential to be improved for learning voice deepfake patterns.

### 2.2. Combining Generative and Predictive Pretraining

Audio generative pretraining focuses on recovering a low-level acoustic representation (e.g., waveform or spectrogram) from latent embeddings or discrete codes (Mousavi et al., 2025; Liu et al., 2023). Intuitively, the acoustic details learned by the generative objective should also benefit discriminative speech tasks. However, recent models that attempted combining predictive and generative pretraining were observed to consistently downperform predictive-only models on discriminative tasks (Liu et al., 2025; Jiang et al., 2025; Mousavi et al., 2025). Missing in previous works is an effective method to integrate generative pretraining without degrading the representation's discriminative power. It is also unclear whether such a pretraining recipe can produce strong representations for various voice deepfake tasks.

## 3. Motivation and Early Exploration

**Motivation.** When building a foundational encoder for voice deepfake tasks, the pretraining objective should help capture traces in deepfakes on both the spoken content and the acoustic details that make the voice sound realistic. The masked token prediction objective, which is commonly used in off-the-shelf SFMs, does not guarantee efficient learning of the acoustic information, as confirmed by previous works on some generation tasks (Hsu et al., 2021; Liu et al., 2025). An supplemental learning objective is needed to capture the acoustic details in the representation. Also, as outlined next, our early explorations reveal that the use of *quantized targets*, as in off-the-shelf pretraining, limits the model's ability to capture deepfake traces in the learned representation.

**Issue with Masked Token Prediction.** We first evaluate the suitability of existing masked token prediction frameworks for voice deepfake tasks. We evaluate two paradigms: the contrastive approach, as exemplified by Wav2vec2, and the predictive approach, as exemplified by HuBERT. We pursue pretraining on Wav2vec2-Base and HuBERT-Base with 1000 hours of real and fake speech, using the same pretraining recipe as the original versions. The scale of pretraining is similar to the original versions pretrained on LibriSpeech-960h (Baevski et al., 2020; Hsu et al., 2021). We then finetune the newly pretrained encoders as well as the original ones for downstream speech deepfake detection, and compare performance. Details of the pretraining, finetuning, and evaluation can be found in Appendix A.

*Table 1.* Equal error rate (EER) performance when comparing encoders pretrained with real+deepfake speech to their original versions. Negative $\Delta$EER suggests performance improvement.

| Encoder | $\Delta$ EER |
|---|---|
| W2V-base (pretrain from scratch) | +1.20 |
| W2V-base (continual pretraining) | +0.80 |
| HuBERT-base (pretrain from scratch) | -0.30 |
| HuBERT-base (continual pretraining) | +0.25 |

In Table. 1, we show the average change in equal error rate (percentage point) with the newly pretrained encoders, when evaluated on 50 speech deepfake detection datasets. Lower EERs and hence negative $\Delta$EERs suggest performance improvement. Given the exposure to 1k hours of fresh data during pretraining, one would expect the new encoders to perform better. However, as shown in Table. 1, with both pretraining from scratch and continual pretraining setups, a slight degradation was seen with all newly pretrained encoders. Across all benchmark datasets, a standard deviation of only 1-2% was seen, suggesting similar effects for different data conditions. These results highlight that adding new data, with the masked token prediction alone, cannot enhance downstream performance on the deepfake detection task.

**Issue with Discrete Pretraining Targets.** To further investigate the drop in performance, one hypothesis we had was that the quantized pseudo-labels do not carry sufficient deepfake-related information. To test this hypothesis, we followed the embedding analysis method proposed in (Hsu et al., 2021), which calculates the mutual information (MI) between the quantized targets and the downstream labels. In Table 2, we report the MI values for two downstream labels, namely deepfake labels and phonetic labels. We note that the original HuBERT has high phonetic-label-conditioned MI values, especially for targets from deeper layers (row 9-11). On the other hand, for HuBERT pretrained with deepfake speech, the deepfake-label-conditioned MI values are relatively much smaller (row 1-6). This observation holds regardless of the feature choice, number of clusters, or the checkpoint iteration. In addition, we experimented with more codebooks by leveraging residual vector quantization (RVQ), a technique commonly used by neural codec models to capture paralinguistic attributes in deeper codebooks (Mousavi et al., 2025). However, no more increase in MI values was seen when scaling beyond two codebooks (row 7-8). These observations confirm that the quantized targets encapsulate very little information related to the deepfake labels. Since the quantized targets guide what information should be learned during pretraining, this helps explain why minimal performance change was seen with the encoders when pretrained with deepfake data. To avoid information loss caused by quantization, we explore the use

of continuous embeddings as pretraining targets.

*Table 2.* Mutual information (MI) values between different pretraining target choices and downstream labels. Note that results in the 'phonetic labels' section are obtained from (Hsu et al., 2021) where pretraining data is LibriSpeech-960h. cls: clusters. it: iteration. RVQ: residual vector quantization.

| No. | Pretraining target choice | MI |
|---|---|---|
| | *— Deepfake labels —* | |
| 1. | MFCCs (100 cls k-means) | .136 |
| 2. | HuBERT-layer6 (500 cls k-means) | .111 |
| 3. | HuBERT-layer6-it2 (500 cls k-means) | .079 |
| 4. | HuBERT-layer6 (2 cls k-means) | .003 |
| 5. | HuBERT-layer6 (100 cls k-means) | .064 |
| 6. | HuBERT-layer6 (1k cls k-means) | .126 |
| 7. | HuBERT-layer6 (1k cls 2-codebook RVQ) | .212 |
| 8. | HuBERT-layer6 (1.5k cls 3-codebook RVQ) | .204 |
| | *— Phonetic labels —* | |
| 9. | MFCCs (100 cls k-means) | .251 |
| 10. | HuBERT-layer6 (100 cls k-means) | .563 |
| 11. | HuBERT-layer6 (500 cls k-means) | .680 |

## 4. Method

The proposed pretraining framework for Alethia is outlined in Figure. 1. The supervision is provided from two branches: (i) a bottleneck masked embedding prediction branch, and (ii) a flow-matching based spectrogram reconstruction branch. Details about the pretraining recipe and data composition are described next.

### 4.1. Pretraining Framework

#### 4.1.1. ENCODER ARCHITECTURE

We pretrain two sizes of encoders, namely *Alethia-Base* with 400M parameters and *Alethia-Large* with 1B parameters. Both include a 7-layer CNN feature extractor followed by a 24-layer and 48-layer transformer encoder for the base and large version, respectively. Other architectural details are described in Appendix B.

#### 4.1.2. MASKING

The student encoder receives input with two types of masks. The first mask is a learnable mask applied to the CNN output, similar to the one proposed for Wav2vec2 (Baevski et al., 2020). Each time step is selected independently at random with a probability of 0.01 to be masked and the mask length is set to 10 time steps. This results in approximately 10% of time steps masked per audio file. We further propose a 2-dimensional mask applied to each transformer layer output. All time steps and feature channels have a probability of 0.15 to be masked, with a maximum of two masks per layer. While the literature has adopted a CNN output mask as the predominant masking strategy (Baevski et al., 2020; Hsu et al., 2021; Chen et al., 2022; Huzaifah et al., 2024), we

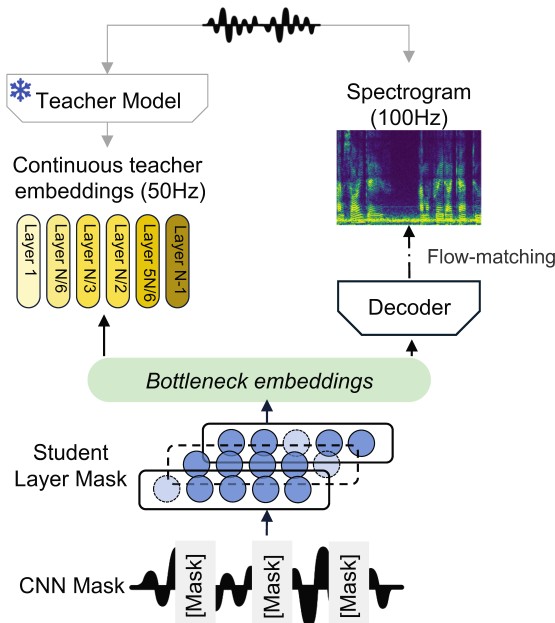

*Figure 1.* Pretraining framework of Alethia. Alethia encoder receives masked waveform and projects it into layer-averaged bottleneck embeddings. These embeddings are then fed in parallel to (1) a *bottleneck masked embedding prediction* branch which predicts different teacher layer embeddings, and (2) a *spectrogram reconstruction* branch to predict the velocity field calculated between the target spectrogram and the decoded spectrogram. The ground-truth embeddings and spectrogram are both obtained from the unmasked waveform.

have found the encoder-layer masking to be crucial for voice deepfake tasks (details can be found in Section 6.6).

### 4.1.3. BOTTLENECK MASKED EMBEDDING PREDICTION

**Predictive Objective.** The objective of this branch is to align the student's bottleneck latent with the frozen teacher's multi-layer feature manifold:

$$\min_{\theta} \mathbb{E}_{\mathbf{x} \sim \mathcal{D}} \left[ \mathcal{L}_{MEP} \left( \{ f_{\text{tea}}^m(\mathbf{x}) \}_{m \in \mathcal{M}}, \Phi(f_{\text{stu}}(\tilde{\mathbf{x}}; \theta)) \right) \right] \quad (1)$$

where $\mathbf{x}$ and $\tilde{\mathbf{x}}$ denote the unmasked and masked waveforms respectively, $\{ f_{\text{tea}}^m(\mathbf{x}) \}_{m \in \mathcal{M}}$ represents the selected $|\mathcal{M}|$ layers of the teacher, $f_{\text{stu}}(\cdot; \theta)$ is the student encoder, and $\mathcal{L}_{MEP}$ represents the embedding prediction loss (Eq. 4). $\Phi$ represents the bottleneck projection which first averages all student layer outputs, then projects into $|\mathcal{M}|$ times higher dimension and reshapes back to $|\mathcal{M}|$ layers to match the teacher's multi-layer embedding dimension. In practice, we select $|\mathcal{M}|$ evenly-spaced layers at different depths from the teacher model. Notably, these raw embeddings are used directly as prediction targets without any discretization to avoid information loss. The teacher model remains frozen throughout the pretraining, while the student encoder and the bottleneck module are trainable.

**Bottleneck Architecture.** Conventionally, distillation aligns the last-layer output of the teacher $f_{tea}^K$ and student $f_{stu}^K$ (Baevski et al., 2023; Assran et al., 2023; 2025). Since our target embeddings are multi-layer, carrying various levels of information from low-level acoustics and speaker attributes to high-level semantic content (Pasad et al., 2021), it can be challenging to compress such rich information into a single layer output. While a layer-to-layer alignment $\{ f_{tea}^m \leftrightarrow f_{stu}^m \}$ (e.g., (Chang et al., 2024)) could make this task easier, it often results in a student model bound by the teacher's performance (Chang et al., 2024; 2025). We therefore design a bottleneck architecture that reconstructs different levels of the teacher's knowledge while avoiding direct copying.

**Loss Function.** The predicted embeddings and target embeddings are aligned using a combination of L1 and cosine distance losses at each layer:

$$\mathcal{L}_{L1} = \sum_{m \in \mathcal{M}} \sum_{t \in \mathcal{T}} \| \mathbf{e}_{m,t} - \hat{\mathbf{e}}_{m,t} \|_1 \quad (2)$$

$$\mathcal{L}_{cos} = \sum_{m \in \mathcal{M}} \sum_{t \in \mathcal{T}} \left( 1 - \frac{\mathbf{e}_{m,t} \cdot \hat{\mathbf{e}}_{m,t}}{\| \mathbf{e}_{m,t} \| \| \hat{\mathbf{e}}_{m,t} \|} \right) \quad (3)$$

$$\mathcal{L}_{MEP} = \alpha \, \mathcal{L}_{L1} + \beta \, \mathcal{L}_{cos} \quad (4)$$

where $\hat{\mathbf{e}}$ and $\mathbf{e}$ are the predicted (student) and target (teacher) embeddings, $\mathcal{M}$ and $\mathcal{T}$ represent selected teacher layers and time steps, and $\alpha$ and $\beta$ are scaling hyperparameters that balance the $L_1$ distance and the cosine distance, respectively. While it is standard practice to calculate losses only at the masked token positions, we have found that with continuous embedding prediction this leads to convergence issues, with increasing loss values at the later training steps (see Appendix C.1). When averaging losses across all time steps (i.e., masked and unmasked positions), training stabilizes with a smooth decrease in the masked position losses.

### 4.1.4. SPECTROGRAM RECONSTRUCTION

The simplest method to reconstruct a low-level acoustic representation $\mathbf{s} \in \mathbb{R}^d$ is a direct mapping via decoding the bottleneck latent as $\hat{\mathbf{s}} = f_\theta(\mathbf{z})$. While some previous works have shown the feasibility of such an approach for distilling smaller students (Guimarães et al., 2024; 2023), we observed significantly higher spectrogram reconstruction error at masked positions compared to unmasked frames. We therefore turn to Flow Matching (Lipman et al., 2022) to learn a probability path conditioned on the bottleneck latent, and find that it leads to similar losses at masked and unmasked positions (Appendix C.2).

**Conditional Flow Matching.** To learn the distribution of sub-perceptual artifacts, we employ Optimal Transport Conditional Flow Matching (OT-CFM) (Lipman et al., 2022). For a minibatch $B$ of clean spectrograms $\{ \mathbf{x}_{0,i} \}_{i=1}^B$ and

noise $\{\mathbf{x}_{1,i}\}_{i=1}^B$ where $\mathbf{x}_{1,i} \sim \mathcal{N}(0, \sigma_{eps}^2 \mathbf{I})$, we first ensure a straight-line probability path by solving for the optimal permutation $p$ using the Hungarian algorithm (Kuhn, 1955) to minimize the quadratic cost $\sum_{i=1}^B \|\mathbf{x}_{0,i} - \mathbf{x}_{1,p(i)}\|^2$. This coupling stabilizes generative pretraining by pairing noise samples with their nearest data neighbors. The state $\mathbf{x}_t$ at time $t \in [0,1]$ and its corresponding ground-truth velocity field $\mathbf{v}_t$ are then defined as:

$$\mathbf{x}_t = t\mathbf{x}_0 + [1 - (1 - \sigma_{min})t]\mathbf{x}_1 \tag{5}$$

$$\mathbf{v}_t = \frac{\mathbf{x}_0 - (1 - \sigma_{min})\mathbf{x}_t}{1 - (1 - \sigma_{min})t} \tag{6}$$

where $\sigma_{min} = 10^{-4}$ ensures numerical stability.

**Generation Objective.** The decoder $g_\psi$ is a transformer that receives the noisy state $\mathbf{x}_t$ (split into real and imaginary components), the time step $t$, and the conditional context $\mathbf{z}_{cond}$ (i.e., the bottleneck embeddings from encoder). The model predicts the real and imaginary velocity fields $\hat{\mathbf{v}}_{real}$ and $\hat{\mathbf{v}}_{imag}$ as:

$$[\hat{\mathbf{v}}_{real}, \hat{\mathbf{v}}_{imag}] = g_\psi(\mathbf{x}_t, t, \mathbf{z}_{cond}) \tag{7}$$

The objective function minimizes the mean squared error (MSE) between the predicted and true velocity fields:

$$\mathcal{L}_{real} = \|\mathbf{v}_{real} - \hat{\mathbf{v}}_{real}\|^2 \tag{8}$$

$$\mathcal{L}_{imag} = \|\mathbf{v}_{imag} - \hat{\mathbf{v}}_{imag}\|^2 \tag{9}$$

$$\mathcal{L}_{FM} = \mathbb{E}_{t,\mathbf{x}_0,\mathbf{z}} \left[ \frac{\mathcal{L}_{real} + \mathcal{L}_{imag}}{\sigma_{eps}^2} \right] \tag{10}$$

The final pretraining loss is a weighted sum of $\mathcal{L}_{FM}$ and the masked embedding prediction loss $\mathcal{L}_{MEP}$ (Equation (4)):

$$\mathcal{L} = \mathcal{L}_{MEP} + \lambda\mathcal{L}_{FM} \tag{11}$$

### 4.2. Pretraining Data

**Audios.** Challenging deepfakes are often found in-the-wild, we therefore target in-the-wild speech as the main component of the pretraining data. Due to the lack of high-quality in-the-wild deepfakes, we self-curated 18k hours of data by leveraging off-the-shelf TTS and VC tools to generate fake voice from CommonVoice (Ardila et al., 2020). We further include 12k hours of public deepfake data from ASVspoof5 (train and dev splits) (Wang et al., 2024), MLAAD (Müller et al., 2024), M-AILABS (Müller et al., 2024), TITW-hard (Jung et al., 2025b), SpoofCeleb (training and dev splits) (Jung et al., 2025a), and ShiftySpeech (Garg et al., 2025). Together, this results in a total of 30k hours of speech data. Details of these datasets can be found in Appendix F. We apply quality control on the accumulated corpus to filter-out poor quality and ineligible audio, resulting in a volume of 19k hours, with real and fake classes well balanced.

**Quality Control.** Since some in-the-wild speech data are of lower quality which may hinder pretraining performance, we filtered the initial 30k-hour pool down to the final 19k hours using a four-stage preprocessing pipeline. First, we performed Voice Activity Detection (VAD) to remove non-speech segments and shorter-duration audios, rejecting samples with less than 1.5s of speech. Second, we employed a speaker diarization model for multi-speaker detection to ensure a clean single-speaker signal. Third, we conducted objective intelligibility estimation, where any audio yielding a Mean Opinion Score (MOS) below 1.5 was rejected. Finally, we applied duration control to ensure that all audios are between 1.5 s and 15 s. Details of the preprocessing modules can be found in Appendix D.

## 5. Experimental Setup

### 5.1. Pretraining Setup

We use WavLM-Large as the teacher for training Alethia-Base and Wav2vec-XLSR-1B for Alethia-Large. Both teachers are frozen throughout pretraining. We select 6 layers, i.e, $|\mathcal{M}| = 6$, from both teacher models, namely $[4, 8, 12, 16, 20, 24]$ from WavLM-Large, and $[4, 12, 20, 28, 36, 42]$ from Wav2vec-XLSR-1B. Pretraining is done for a total of 600k and 300k steps respectively, equivalent to one epoch for the base and large models. Hyperparameters of the loss functions (Equation (4) and Equation (11)) are set as: $\lambda = 0.25$, $\alpha = 1$, $\beta = 1$. Other details can be found in Appendix B.

### 5.2. Downstream Finetuning Setup

We evaluate Alethia against four commonly used SFMs for learning tasks on deepfakes (Li et al., 2025a), namely Wav2vec-XLSR-300M (referred to as W2V-300M hereinafter), WavLM-Large, HuBERT-Large, and Wav2vec-XLSR-1B (referred to as W2V-1B hereinafter). The former three have similar size as Alethia-Base and W2V-1B has the same size as Alethia-Large.

Evaluations are conducted on a total of 5 learning tasks, namely speech deepfake detection (SDD), singing voice deepfake detection (SVDD), partially fake speech localization (PFSL), source tracing (ST), and audio-visual deepfake detection (AVDD), including a total of 56 datasets. To the best of our knowledge, this is the largest voice deepfake benchmark at the time of writing. For utterance-level tasks, we report equal error rate (EER) as the main metric, supplemented by accuracy (Acc), true positive rate (TPR), and true negative rate (TNR) computed with a fixed threshold. For frame-level tasks, frame-wise EER is reported. Details of task-specific finetuning and benchmarking can be found in Appendix E and F.

*Table 3.* Summary of SDD finetuning setups. Detailed data composition of the three setups can be found in Appendix F. RB: RawBoost. Perturb: a group of perturbation augmentations. Voc: vocoder augmentation.

| Setup | Train data | Augmentation |
|---|---|---|
| LOW-RESOURCE | SDD-FT-400 | RB |
| EXPANDED | SDD-FT-3k | RB |
| EXPANDED+AUG | SDD-FT-12k | RB+Perturb+Voc |

## 6. Experiments

### 6.1. Speech Deepfake Detection (SDD)

**Architecture.** The same downstream classifier architecture is applied to all encoders, including an average pooling along layer and time dimensions, followed by a 2-layer MLP with ReLU activation and dropout. Models are fully-finetuned, i.e., no frozen components, with binary cross-entropy loss. We perform hyperparameter tuning for each model using grid search and train for a maximum of 5 epochs.

**Setup and Datasets.** We design three finetuning setups summarized in Table. 3. The LOW-RESOURCE setup simulates how most existing detection models are trained, where limited training data (i.e., 400h) is provided and is augmented by RawBoost (Tak et al., 2022). In the EXPANDED setup, we increase the finetuning data to 3.3k hours, consisting of public datasets used in the LOW-RESOURCE setup together with self-curated data to maximize model performance. In the EXPANDED+AUG setup, we further add various signal perturbation types as well as their randomized combinations, together with 10 types of vocoder augmentations to probe the upper limit of finetuning benefits. For evaluation, we compose a large SDD benchmark of 50 datasets (referred to as 'SDD-Eval-50' hereinafter) with a wide coverage of different speaking styles, languages, accents, perturbations, room acoustics, and generative models.

**Results.** We report average metrics obtained on SDD-Eval-50 in Table 4 and defer the per-dataset performance comparison to Appendix G.1. We note an overall increase in average accuracy as the the training data volume is scaled. However, despite expanding the finetuning dataset to 12k hours (EXPANDED+AUG setup), general-purpose SFMs exhibit a persistent lack of generalization on some challenging datasets. Take the best baseline model W2V-1B as an example. W2V-1B achieves a high average accuracy of 91.9%, but the mean value masks the high variance within the eval datasets where 17 out of 50 datasets yield accuracies below 90%, and 6 datasets yield below 80% (details in Appendix G.1). The lack of generalization is particularly prominent for in-the-wild deepfakes, such as in Deepfake_eval_2024 (Chandra et al., 2025), and for audios altered with acoustic perturbations, such as in ReplayDF (Müller

*Table 4.* SDD model performance comparison. For the three finetuning setups, we report average metrics across 50 evaluation datasets. *Challenging* datasets vary per condition, which refer to the subset where the EER from W2V-1B is worse (higher) than the mean across all datasets.

| Model | All | | Challenging | |
|---|---|---|---|---|
| | EER ↓ | Acc ↑ | EER ↓ | Acc ↑ |
| HuBERT-Large | 11.4 | 84.0 | 18.7 | 73.6 |
| WavLM-Large | 8.0 | 85.9 | 15.0 | 74.5 |
| W2V-300M | 14.1 | 71.8 | 21.1 | 61.3 |
| W2V-1B | 6.0 | 91.9 | 13.2 | 78.2 |
| Alethia-Base | 6.9 | 90.6 | 13.1 | 80.7 |
| Alethia-Large | **5.2** | **93.3** | **11.5** | **81.2** |

et al., 2025) and FoR-rerecorded (Reimao & Tzerpos, 2019). Clearly, scaling finetuning data alone is not sufficient to achieve a high degree of generalization.

When comparing Alethia to the other SFMs, we see consistently better EER and accuracy across all three finetuning setups. Notably, the performance gains are most pronounced on the challenging subset: under the EXPANDED setup, when compared to W2V-1B, Alethia-Large achieves a 2.2% reduction in EER and a 6.4% increase in accuracy. Similar trend is seen under the EXPANDED-AUG setup as well. Furthermore, Alethia demonstrates smaller performance variance across different evaluation datasets. With Alethia-Large (EXPANDED-AUG condition), only 11 datasets fall below 90% accuracy and 4 datasets below 80% accuracy (details in Appendix G.1). Additionally, Alethia-Base demonstrates comparable performance with W2V-1B and markedly better performance than other SFMs at similar sizes. Results here demonstrate the superior generalizability and robustness of Alethia compared to existing SFMs.

### 6.2. Singing Voice Deepfake Detection (SVDD)

**Setup.** As a newly emerged learning task, SVDD currently lacks the dataset scale of the SDD task. Standard evaluation practice for this task is to use a non-overlapping split from the same dataset on which the model is trained (Zang et al., 2024). A foundational encoder for voice deepfakes should exhibit zero-shot generalization to singing voice deepfakes, due to the shared physiological basis of vocalization in speech and singing voice. We evaluate Alethia checkpoints, finetuned exclusively with SDD datasets under the EXPANDED-AUG setup, on the CtrSVDD test split (Zang et al., 2024). The training and validation sets of CtrSVDD remain unseen during all stages, providing a stringent evaluation of the model's zero-shot generalizability.

**Results.** From Table 5, we note that both Alethia-Base and Alethia-Large exhibit superior zero-shot generalization compared to other SFMs of similar size, achieving EERs of 16.7% and 10.8%, respectively. Notably, Alethia-1B

*Table 5.* Model performance on the SVDD task. CtrSVDD baseline is finetuned solely on in-domain singing voice data. (* as reported by the original authors.)

| Model | EER ↓ | Acc ↑ | TPR ↑ | TNR ↑ |
|---|---|---|---|---|
| *Zero-shot testing* | | | | |
| HuBERT-Large | 25.7 | 88.9 | 97.3 | 39.7 |
| WavLM-Large | 22.6 | 89.8 | **97.7** | 43.5 |
| W2V-300M | 22.4 | 84.9 | 88.6 | 63.7 |
| W2V-1B | 13.2 | 89.7 | 90.8 | 83.1 |
| Alethia-Base | 16.7 | 89.8 | 94.0 | 65.2 |
| Alethia-Large | **10.8** | **91.3** | 92.5 | **84.1** |
| *In-domain finetuning* (Zhang et al., 2024) | | | | |
| CtrSVDD Baseline* | 13.8 | – | – | – |

*Table 6.* Model performance on the PFSL task, as measured with 20 ms frame-level EERs. PS: PartialSpoof. HT: Half-Truth. LPS: LlamaPartialSpoof. (* as reported by the original authors.)

| Model | Dataset | | | Average |
|---|---|---|---|---|
| | PS | HT | LPS | |
| *Zero-shot testing* | | | | |
| HuBERT-Large | 29.9 | 32.9 | 24.0 | 28.9 |
| WavLM-Large | 28.0 | 18.1 | 19.0 | 21.9 |
| W2V-300M | 28.0 | 31.5 | 21.8 | 27.1 |
| W2V-1B | 25.4 | 14.2 | 20.7 | 20.1 |
| Alethia-Base | 27.1 | 9.2 | 23.2 | 19.8 |
| Alethia-Large | 27.2 | 10.0 | 19.8 | **19.0** |
| *In-domain finetuning* (Luong et al., 2025) | | | | |
| PS-finetuned Baseline* | 13.7 | 46.4 | 46.3 | 35.5 |
| HT-finetuned Baseline* | 49.5 | 0.1 | 59.4 | 36.3 |

outperforms the CtrSVDD baseline by a margin of 3.0% in EER, despite the latter being explicitly optimized on in-domain training and development sets. The performance gains from Alethia underscore the efficacy of the proposed pretraining recipe in capturing generation artifacts invariant to vocal type, i.e., whether standard speech or singing voice.

### 6.3. Partially Fake Speech Localization (PFSL)

**Setup.** The partially fake localization task typically requires frame-level supervision, with models finetuned at high temporal resolutions (e.g., 20 ms) to capture local synthetic boundaries. We investigate whether the representations learned during large-scale SDD finetuning are sufficient to facilitate localization without further task-specific finetuning. To test this, we evaluate our SDD-finetuned models in a zero-shot manner by removing the temporal pooling layer and applying the utterance-level classifier to the intermediate embeddings at each time step. This protocol directly probes the granularity of the deepfake artifacts encapsulated within the encoder's latent space. Evaluations are conducted on PartialSpoof (Zhang et al., 2022), Half-Truth (Yi et al., 2021), and LlamaPartialSpoof (Luong et al., 2025) at 20 ms resolution.

**Results.** From Table 6, we note both Alethia-Base and Alethia-Large outperform other SFMs of comparable size. Notably, while models finetuned directly on PartialSpoof or Half-Truth achieve top performance on their respective in-domain test sets, their generalizability to unseen datasets degrades to near-random chance. In contrast, the representations learned by Alethia through large-scale SDD finetuning exhibit much better generalization under zero-shot testing. Overall, Alethia not only provides superior global representations, as demonstrated in the SDD task, but also effectively encapsulates fine-grained synthetic artifacts at the frame level, as demonstrated in the PFSL task.

### 6.4. Source Tracing (ST)

**Setup.** The goal of ST is to infer the category of generation method for deepfake samples. We consider pretrained speech encoders with ∼300M parameters, and compare their performance in two ways. First, we extract final-layer embeddings from each encoder following Xuan et al. (2025), then assess their ability to separate source categories without task-specific fine-tuning. Embedding separability is quantified using the silhouette score $\in [-1, 1]$ (Shahapure & Nicholas, 2020), computed directly in the original embedding space with cosine distance. Second, we pursue downstream classification with an MLP classifier attached to the frozen encoder and trained via cross-entropy loss (details in Appendix E.2). We work with a random 10% subset of the ASVspoof5 (Wang et al., 2024) *eval* set, amounting to 68k samples and denoted as ASVspoof5-ST. This subset contains 17 source classes (including bonafide) and is completely unseen during pretraining. We partition ASVspoof5-ST into train (80%), validation (10%) and test (10%) splits. The test split is used for evaluating both the embedding separability and downstream classification.

**Results.** As seen in Table 7, Alethia-Base achieves a higher silhouette-score than other encoders, indicating stronger source tracing capability. Qualitative UMAP visualizations of the embedding spaces for both the settings are provided in Appendix G.2. When finetuning is pursued with frozen encoders, Alethia-Base substantially outperforms other encoders, with 98.8% test accuracy, suggesting that the embeddings encode rich source-discriminative information.

### 6.5. Audio-Visual Deepfake Detection (AVDD)

**Setup.** In the AVDD task, we evaluate the performance of SFMs on audio-visual deepfake datasets, where one or both of the audio and visual modalities are manipulated. Methods including TTS, voice conversion, face swapping, and lip-syncing are commonly applied to create realistic deepfake

*Table 7.* Model performance on the ST task. Higher Silhouette scores indicate better embedding separability.

| Model | Embedding Separability | MLP finetuning |
|---|---|---|
| | SilhouetteScore ↑ | Acc ↑ |
| HuBERT-Large | −0.10 | 51.0 |
| WavLM-Large | −0.03 | 78.9 |
| W2V-300M | −0.15 | 26.9 |
| Alethia-Base | **+0.02** | **98.8** |

*Table 8.* EER comparison for the AVDD task.

| Model | Dataset | | Average |
|---|---|---|---|
| | FakeAVCeleb | PolyGlotFake | |
| *Zero-shot testing* | | | |
| HuBERT-Large | 8.1 | 13.9 | 11.0 |
| WavLM-Large | 7.0 | 14.1 | 10.6 |
| W2V-300M | **5.8** | 9.4 | 7.6 |
| Alethia-Base | 6.3 | **7.1** | **6.7** |

videos. We evaluate all models, finetuned for the SDD task under the EXPANDED condition, in a zero-shot manner on two audio-visual deepfake datasets: FakeAVCeleb (Khalid et al., 2021) and PolyGlotFake (Hou et al., 2024). Both datasets contain three categories of labeled deepfakes (audio only, video only, and both), with PolyGlotFake employing more recent and challenging deepfake techniques. Additional details on the dataset pre-processing and setup can be found in Appendix E.2.

**Results.** We compare the EER performance of Alethia-Base with other encoders of similar size in Table 8. All the models have been finetuned for the SDD task. We observe that Alethia has consistently smaller EER, particularly on the more challenging PolyGlotFake. We also report additional metrics including accuracy, TPR, and TNR in Table 18 in the Appendix, and observe a similar trend.

## 6.6. Ablations

We highlight the importance of key design choices, while deferring other ablation experiments to Appendix C.

**Generative Objective.** We evaluate the impact of the generative component by pretraining an Alethia-Base variant using only the bottleneck masked embedding prediction objective. As shown in Table 9, removing the generative branch results in an overall performance drop ($\Delta$EER $= +0.5$), primarily driven by a significant reduction in sensitivity ($\Delta$TPR $= -2.5$). Notably, the generative objective is critical for capturing artifacts from modern high-fidelity synthesizers; for attacks based on diffusion and flow-matching, the generative branch yields a substantial $\Delta$EER improvement of 5.6. Conversely, performance on more traditional generative methods remains nearly static ($\Delta$EER $= -0.1$). These findings suggest that generative pretraining provides

*Table 9.* Change in SDD performance after removing the generative objective during pretraining. Degradations are highlighted in red. Diff: Diffusion generated deepfakes. FM: Flow-matching generated deepfakes.

| Data Category | w/o generative objective | | | |
|---|---|---|---|---|
| | $\Delta$EER | $\Delta$Acc | $\Delta$TPR | $\Delta$TNR |
| All | +0.5 | −0.7 | −2.5 | +2.0 |
| Diff & FM | +5.6 | −2.2 | −3.0 | +0.3 |
| Others | −0.1 | −0.6 | −2.5 | +2.2 |

an inductive bias essential for capturing traces of different voice-synthesis methods.

**Bottleneck Architecture.** To isolate the contribution of the bottleneck design from the masking pretext task, we performed an ablation study using an Alethia-Base variant. In this configuration, we removed both the masking procedure and the generative flow-matching objective, effectively reducing the task to a simple student-teacher reconstruction of the teacher's multi-layer manifold from the unmasked student latent. We found that even in the absence of masking, the bottleneck architecture alone provides an EER improvement of 0.7% and an accuracy gain of 0.49% over the WavLM-Large baseline. These results suggest that the bottleneck itself serves as a powerful architectural prior for the pretraining objective.

**Masking Methods.** We experimented a variety of masking method setups. Due to the compute limit, these experiments were performed only with Alethia-Base. We follow the EXPANDED condition for downstream finetuning and select a subset of 30 benchmark datasets from the SDD-Eval-50 for model evaluation, as the full benchmarking run takes 24h on 8 H100 GPUs. Table 10 summarizes the performance variation compared to the default configuration. When comparing different CNN masking types with the same masking ratio of 10%, learnable mask is found the most effective with $1-2\%$ improvement in EER. We also attempted using other perturbation combinations sampled from the self-curated 'AugmentedDeepfake-2k' dataset instead of gaussian noise, but found this lead to non-convergence of pretraining, likely due to the lower quality of audios after augmentation. Regarding masking ratios, we started with the 50% global masking percentage which is close to the setup of general-purpose SFMs (Baevski et al., 2020; Chen et al., 2022; Hsu et al., 2021). We found that for our tasks, 10% of masking leads to better performance than higher ratios. This could be because our pretraining data are dominated by in-the-wild speech, the noise level of which is likely higher than the pretraining data used for general-purpose SFMs (Baevski et al., 2020; Chen et al., 2022; Hsu et al., 2021). As a result, prediction of masked frames becomes more challenging, hence favoring a lower masking ratio. Lastly, a degradation of 2.3% in EER is observed when removing the student

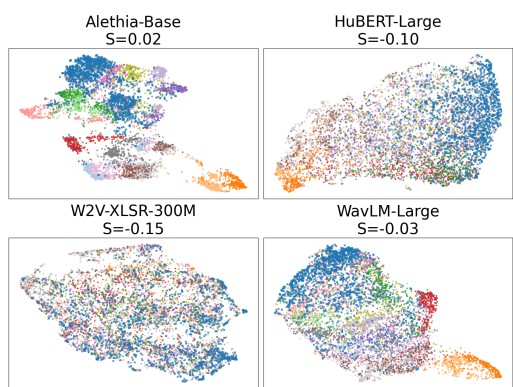

*Figure 2.* UMAP projections of pretrained model embeddings without any task-specific fine-tuning, colored by source labels for the ASVspoof5-ST test split, where $S$ denotes the Silhouette Score.

layer mask.

*Table 10.* Masking ablation results. LM: Student layer mask. NC: Non-convergence.

| ID | Mask Type | Ratio | LM | EER |
|---|---|---|---|---|
| Default | Learnable | 10% | ✓ | 16.2 |
| 1 | Learnable | 20% | ✓ | 19.2 (+3) |
| 2 | Learnable | 30% | ✓ | 19.3 (+3.1) |
| 3 | Learnable | 50% | ✓ | NC |
| 4 | Zero | 10% | ✓ | 17.4 (+1.2) |
| 5 | Gaussian Noise | 10% | ✓ | 18.9 (+1.5) |
| 6 | Perturbations | 10% | ✓ | NC |
| 7 | Learnable | 10% | ✗ | 18.5 (+2.3) |

## 7. Conclusion

We propose *Alethia*, the first foundational encoder that generalizes across a variety of voice deepfake tasks. By combining a bottleneck *masked-embedding prediction* task with a *generative flow-matching* objective, Alethia learns to capture deepfake generation artifacts in a self-supervised manner. Extensive evaluations across 5 tasks on 56 datasets reveal that Alethia outperforms SOTA foundation models under identical finetuning conditions. Alethia also exhibits robust zero-shot generalization to unseen deepfake domains, such as singing voice and partially fake speech.

## Impact Statement

As high-quality deepfakes become easier to create for spreading misinformation or committing fraud, having a scalable way to defend against these attacks is vital. By proposing a novel foundation model and pretraining recipe that generalizes across a broad scope of deepfake tasks, our research helps improve the safety of digital communications. While the arms race between deepfake generation and detection will continue, our work offers a principled way to help prevent and combat the misuse of AI-generated content.

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

## A. POC Experiment Setup

We follow a similar setup as the base versions of Wave2vec2 and HuBERT, where the pretraining data are identical to the downstream finetuning data (Baevski et al., 2020; Hsu et al., 2021). We curate 1k hours of real and fake speech, where the real speech data are sourced from CommonVoice (Ardila et al., 2020) and the fake speech data are generated using 20 different off-the-shelf generative models. With both models, we perform two types of pretraining, including (1) continue pretraining on self-curated data with the original LibriSpeech-pretrained checkpoint as the *warm start*, and (2) pretrain *from scratch* on self-curated data. We reuse the default hyperparameters set in the original pretraining recipes for our experiments. With finetuning, we again use the labeled 1k hours of data and append a 2-layer MLP with dropout of 0.5 to all encoders. We employ the AdamW optimizer with a base learning rate of $4 \times 10^{-4}$ and weight decay of $1 \times 10^{-4}$. We adopt a cosine annealing scheduler with a minimum learning rate of $4 \times 10^{-5}$ and a start factor of 0.01. The learning rate is scaled linearly with the number of GPUs. For benchmarking, we use the SDD datasets listed in Table 15.

## B. Alethia Architecture and Pretraining Details

Table 11 summarizes the architectural hyperparameters of Alethia-Base and Alethia-Large. One thing to note is that while the teacher model WavLM-Large adopts PostLN, the student Alethia-Base uses PreLN, with which we found better convergence. For both base and large versions, we experimented with random initialization and initialization from pre-trained checkpoints. With the 400M version, the pre-trained initialization leads to slightly better performance (0.5% EER difference across the 56 datasets). With the 1B version, the average EER difference is not significant (¡0.1% EER).

Regarding pretraining setup, we utilize the AdamW optimizer with a learning rate schedule with a linear warm-up proportion of 0.07, reaching a peak learning rate of $2 \times 10^{-4}$. The optimizer hyperparameters are configured with $\beta_1 = 0.9$, $\beta_2 = 0.98$, and $\epsilon = 10^{-6}$, applying a weight decay of $10^{-6}$. Experiments are done on $8 \times$ NVIDIA H100 GPUs with a per-GPU batch size of 12 for Large and 28 for Base, corresponding to approximately one epoch of pretraining for both sizes.

## C. Ablations on Other Pretraining Components

### C.1. Effect of Global Supervision on Predictive Objective

We empirically observed that calculating $\mathcal{L}_{MEP}$ over all time steps $\mathcal{T}$, rather than solely at masked positions, improves convergence. As is shown in Figure. 3, this "all-step" supervision leads to lower prediction error even when evaluated exclusively on masked frames. We attribute this to the fact that unmasked positions act as reference anchors, which regularizes predictive loss and prevents the student's projection head $\Phi$ from diverging from the teacher's manifold.

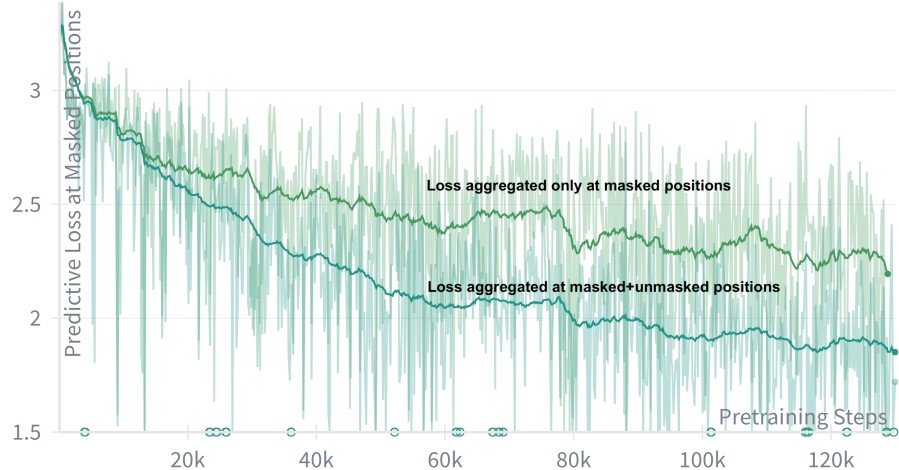

*Figure 3.* Masked embedding prediction error calculated at masked time steps. The global supervision leads to lower prediction error even when evaluated exclusively on masked frames.

*Table 11.* Architectural Hyperparameters for Alethia Pretraining Configuration.

| Category | Hyperparameter | Alethia-Large | Alethia-Base |
|---|---|---|---|
| **Encoder** | CNN Layers | [(512,10,5)] + [(512,3,2)] * 4 + [(512,2,2)] * 2 | |
| | CNN Dropout | 0 | 0 |
| | Convolution Bias | True | True |
| | Transformer Layers | 48 | 24 |
| | Embedding Dimension ($D$) | 1280 | 1024 |
| | FFN Inner Dimension | 5120 | 4096 |
| | Attention Heads | 16 | 16 |
| | Positional Embedding (Conv) | 128 (Groups=16) | 128 (Groups=16) |
| | Dropout (Hidden / Attention) | 0.1 / 0.1 | 0.1 / 0.1 |
| | LayerNorm | PreLN | PreLN |
| **Bottleneck** | Target Teacher Model | Wav2vec2-XLSR-1B | WavLM-Large |
| | Teacher Target Layers ($\mathcal{L}$) | [4, 12, 20, 28, 36, 41] | [4, 8, 12, 16, 20, 24] |
| | Bottleneck Projection Type ($\Phi$) | Average Pooling | Average Pooling |
| | Loss Components | $L_1$ + Cosine Similarity | $L_1$ + Cosine Similarity |
| **Generative Head** | Decoder Layers | 8 | 4 |
| | Hidden / Time Embedding Dim | 1280 | 1024 |
| | Attention Heads | 16 | 16 |
| | FFN Dimension | 2560 | 2048 |
| | Flow-Matching std ($\epsilon$) | 2.0 | 2.0 |
| | Reconstruction Weight ($\lambda$) | 0.25 | 0.25 |
| **Spectrogram** | Number of FFT Points | 512 | |
| | Hop Length | 160 | |
| | Window Length | 400 | |
| **CNN Masking** | Masking Mode | Learnable Mask | |
| | Mask Probability | 0.10 | |
| | Mask Length | 10 Steps | |
| | Time Masking Prob | 0.15 | |
| **Student Layer Masking** | Masking Mode | Zero Mask | |
| | Feature Mask Probability | 0.15 | |
| | Feature Mask Number | 2 | |
| | Feature Mask Percentage | 0.15 | |
| | Time Mask Probability | 0.15 | |
| | Time Mask Number | 2 | |
| | Time Mask Percentage | 0.15 | |

### C.2. Spectrogram Reconstruction Methods

We investigate the spectrogram reconstruction ability of different decoding methods by comparing the generative loss at masked, unmasked, and all frame positions. Since the learnable mask is applied to the CNN output and there is a resolution gap between the CNN output and the spectrogram, we first map the masked CNN token positions back to waveform domain using the window length and stride specified in Table B, we then estimate the positions of masked spectrogram frames with the pre-defined FFT parameters. In cases where a portion of the frame is masked, we consider them as masked. Figure. 4 demonstrates how the generative loss differs when different decoding methods are used. When the spectrogram is generated directly with a decoder (i.e., regression), the model tends to overfit on unmasked frames where the loss at masked frames is not reduced. With flow-matching, similar loss is seen across all frame positions with smoother convergence.

## D. Preprocessing Modules

Table 12 summarizes the hyperparameters of preprocessing modules used for quality control. For VAD, we employ `Silero VAD v5` with a threshold of 0.5, a minimum speech duration of 50 ms, a silence duration of 50 ms, and a speech padding of 180 ms. Multi-speaker instances are identified via `pyannote-audio v3.1.1` speaker diarization, where we reject any segment containing excessive overlapping speech or significant secondary speaker presence. Lastly, we assess objective speech quality using `DNSMOSPro`, discarding samples with a MOS score below 1.5 to eliminate unintelligible recordings.

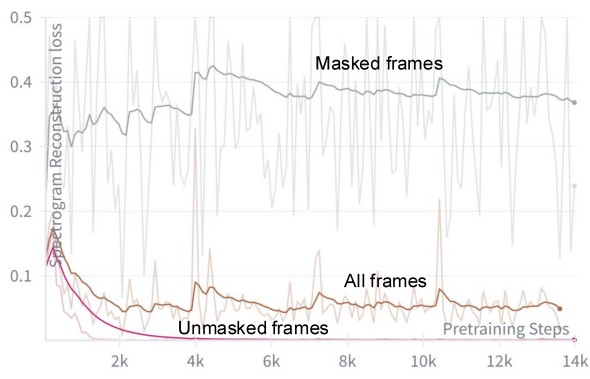 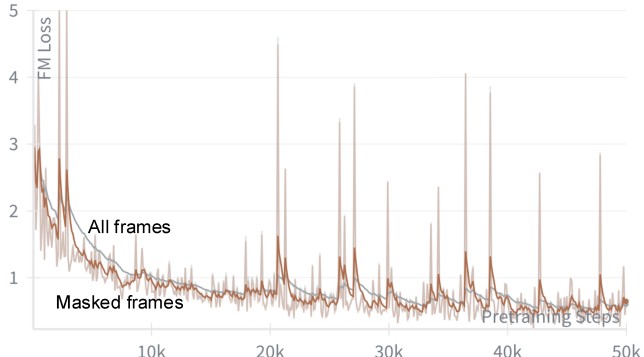

*Figure 4.* Left: Spectrogram reconstruction loss measured by comparing groundtruth spectrogram with the spectrogram generated directly with a decoder (i.e., regression). Right: Flow-Matching (FM) velocity field prediction loss where the velocity prediction is conditioned on the decoder output. The regression method tends to overfit reconstruction on unmasked frames, while the FM method leads to similar loss across all frames.

*Table 12.* Hyperparameters and rejection criteria for quality control modules.

| Module | Model | Rejection Threshold |
|---|---|---|
| VAD | Silero VAD v5 | Total speech duration $< 1.5$s |
| Multi-Speaker | pyannote-audio v3.1.1 | Overlap $> 1$s or secondary speaker $> 1.5$s |
| Intelligibility | DNSMOSPro | Mean Opinion Score (MOS) $< 1.5$ |
| Duration | Librosa | Length $\notin [1.5s, 15s]$ |

## E. Downstream Task Setup Details

### E.1. SDD and SVDD

**Preprocessing.** Waveforms are resampled to 16kHz and unified to 3s with circular padding. Peak amplitude normalization is performed to standardize the volume of each audio file.

**Classifier.** The 3D embeddings extracted by the frontend are pooled along layer and time axes using average pooling. The output embeddings are then processed by a 2-layer MLP with 1280 and 16 input neurons, with a dropout of 0.5 for both layers. The final logit output is converted into a probability score between 0 and 1 using sigmoid function.

**Finetuning and Evaluation Setup.** With LOW-RESROUCE and EXPANDED conditions, a global batch size of 1568 is used to train 300M models and 768 to train 1B models. With the EXPANDED-AUG condition, we use a reduced global batch size of 256 to account for extra VRAM needed for online vocoder augmentation. Models are optimized using AdamW with a base learning rate (LR) of $4 \times 10^{-6}$ and a weight decay of $10^{-4}$ for 1B models, and a base LR of $4 \times 10^{-5}$ with the same decay for 300M models. A cosine annealing scheduler with a minimum LR of $4 \times 10^{-7}$ is employed for all models. The schedule begins with a start factor of $0.01$ and utilizes a warmup factor of $0.1$ to ensure numerical stability during the initial phase of training. With the above setup, we observed similar training and validation losses for models at similar sizes, with relatively slightly faster convergence seen with Alethia. Each finetuning run takes about 16, 96, and 800 GPU hours for LOW-RESOURCE, EXPANDED, and EXPANDED-AUG, respectively. With evaluation, we use a global batch size of 8064 with full precision, which leads to 88 GPU hours of inference time for the full benchmarking run. Note that different datasets may have contradictory labels for generated audios, for example, the vocoded speech are labeled as fake by CVoiceFake while treated as real by ASVspoof5. We therefore unified labels of these special categories of deepfakes. These label handling details can be found in the 'Notes' column in Table 15.

### E.2. Other Tasks

**PFSL.** For dataset configuration and model inference, we adapt the framework provided by Luong et al. (2025).[1]. While the original framework supports multi-scale evaluation across six temporal resolutions (units $\in$

---

[1] https://github.com/hieuthi/MultiResoModel-Simple

$\{0.02, 0.04, 0.08, 0.16, 0.32, 0.64\}$ seconds), we evaluate exclusively at the 20ms scale (units = 0.02). This choice is motivated by the fact that the finest temporal resolution represents the most challenging detection scenario and provides the most precise localization of synthetic boundaries (Zhang et al., 2022; Luong et al., 2025). The evaluation is performed with SDD model checkpoints without retraining. To bridge the resolution gap, we remove the temporal pooling operation which reverts the global embeddings of shape $(\mathcal{B}, \mathcal{F})$ back to $20\,\text{ms}$-based embeddings with shape of $(\mathcal{B}, \mathcal{T}, \mathcal{F})$. The pretrained classifier is then applied to each time step to obtain a per-frame decision.

**ST.** Similar to Xuan et al. (2025), all input audio is resampled to 16kHz and normalized to 4s using circular padding. Final layer encoder embeddings are average-pooled over time axis and used directly for calculating silhouette scores for pretrained embedding evaluation. For downstream evaluation, pooled embeddings are passed to a 2-layer MLP with 1024 and 256 input neurons and LeakyReLU (slope 0.1) activation, where only the MLP is trained while the encoder remains frozen. Training is performed on a random 10% subset of the ASVspoof5 evaluation dataset, denoted as ASVspoof5-ST, which is further split into $80/10/10$ train/validation/test splits. Following the training setup provided by Xuan et al. (2025), we optimize cross-entropy loss using Adam with a learning rate of $5 \times 10^{-4}$, $\beta_1 = 0.9$, $\beta_2 = 0.999$, $\epsilon = 10^{-8}$, and a weight decay of $5 \times 10^{-4}$ with a batch size of 84 for 50 epochs. The learning rate is decayed by 0.5 every 10 epochs, and we use mixup, where the mixing ratio $\lambda$ is drawn from a Beta distribution with parameters $\alpha = 0.5$. We omit results for full encoder fine-tuning with an MLP classifier, since this setting resulted in uniformly saturated performance across all encoders and did not yield meaningful distinctions for comparison. Additionally, we do not report results on MCL-MLAAD (Xuan et al., 2025), as MLAAD (Müller et al., 2024) is included in the pretraining of Alethia, which could lead to biased evaluation.

**AVDD.** For this task, we adapt code from the FakeAVCeleb repository [2]. We segmented the videos into 3 second chunks to handle variable durations, and treat the chunks as independent samples while calculating metrics. The audio stream is extracted from the videos and resampled to 16 kHz, PCM wav format. In the FakeAVCeleb paper (Khalid et al., 2021), they extract image and audio features at a frame level (25 ms), and for audio they use Mel-Spectrograms resized into 3-channel images as input to an identical detector as the image (video) inputs. Since Alethia and the other foundation models we use all take the raw audio samples as input, we follow a different approach and obtain a single prediction for 3-second audio samples. For the video inputs, we uniformly sample a fixed number of 25 frames from the 3-second chunks, and combine the frame-level predictions using a Sigmoid applied to the average logits corresponding to the frames. This gives a real or fake prediction for the 3-second video samples.

## F. Datasets

**Pretraining.** The composition of the Alethia pretraining corpus is detailed in Table 13. To align the pretraining distribution with the typical duration of downstream detection tasks, we segmented longer recordings from the Mix-10k, ASVspoof5, and MLAAD subsets into 3s and 6s chunks with 50% overlap, prior to quality control. To simulate real-world deepfake characteristics, we prioritized samples carrying in-the-wild acoustic characteristics, such as media compression and varying microphone responses. For similar purposes, we emphasize speech collected from uncontrolled environment, including both scripted and spontaneous speech, to ensure the encoder is robust to the stylistic variations typical of production-level deepfakes.

**Finetuning.** SDD finetuning data composition can be found in Table 3. Mix-10k is our self-curated dataset with pairs of real and deepfake speech generated from a variety of academic and commercial voice generation models, such as Tortoise, Tacotron, VITS, ElevenLabs, Cartesia, PlayHT, etc. The other six datasets are publicly available. For all experimental conditions, we performed quality control and class balancing for each training dataset. While this significantly reduces the total training volume, it helps to avoid the effects of confounding factors, such as prolonged silence and class imbalance, which may otherwise lead to spurious correlations (Müller et al., 2021). For the other tasks, we kept the training data in their original state.

**Evaluation Benchmarks.** A comprehensive list of evaluation datasets is provided in Table 15. Datasets subjected to our quality control pipeline are denoted with the "preprocessed" suffix, whereas those labeled "raw" remain in their original state. To facilitate a direct comparison with prior literature, we specifically utilize the raw versions of standard benchmarks, including the ASVspoof series (Nautsch et al., 2021; Yamagishi et al., 2021; Wang et al., 2024), in-the-wild (Müller et al., 2022), CtrSVDD (Zang et al., 2024), and datasets used for PFSL and AVDD tasks. This allows us to validate Alethia against historical baselines on individual datasets, while simultaneously assessing its performance on preprocessed data with

---

[2] https://github.com/DASH-Lab/FakeAVCeleb

*Table 13.* Summary of Alethia pretraining data composition after quality control.

| Dataset | Hrs ($10^3$) | Perturbation | Contain in-the-wild speech |
|---|---|---|---|
| Mix-10k | 10.2 | Compression + microphone artifacts | Yes |
| ASVspoof5 (tr+dev) | 1.1 | Compression + adversarial noise | NA |
| MLAAD | 0.73 | NA | NA |
| M-AILABS | 0.99 | NA | NA |
| TITW-hard | 0.15 | May contain Youtube compression artifacts | Yes |
| SpoofCeleb (tr+dev) | 1.81 | May contain Youtube compression artifacts | Yes |
| ShiftySpeech | 4.2 | May contain Youtube compression artifacts | Yes |
| **Total** | **19.18** | Various | Yes |

*Table 14.* Finetuning data composition per SDD condition.

| SDD Training Data | #Samples | Dataset | Augmentation |
|---|---|---|---|
| SDD-FT-400 | 459,361 | ASV19, ASV5, In-the-wild | RawBoost |
| SDD-FT-3k | 3,938,116 | ASV19, ASV5, In-the-wild, Self-curated data | RawBoost |
| SDD-FT-14k | 16,553,908 | ASV19, ASV5, In-the-wild, Self-curated data | RawBoost + Perturbations + Vocoder |

better integrity. While a strictly zero model overlap is difficult and cumbersome to achieve when running evaluations on a large scale, we have explicitly avoided using code or hyper-parameters provided in any of the evaluation repositories. By focusing on evaluations at scale, our goal is to assess whether Alethia learns generalizable generative artifacts rather than overfitting to certain dataset-specific model signatures. Among the 56 evaluation datasets, those that have explicit overlap with pretraining data, albeit without ground-truth labels, have been duly demarcated in Table 15.

*Table 15.* Evaluation datasets categorized by tasks. Datasets subjected to our quality control pipeline are denoted with the "preprocessed" suffix, whereas those labeled "raw" remain in their original state. Datasets are ordered alphabetically.

| Task | Dataset | #Samples | Notes |
|---|---|---|---|
| **SDD** | ALM-ADD_raw (Xie et al., 2024) | 357 | Audios without speech/voice excluded |
| | ASVspoof2019_LA_raw (Nautsch et al., 2021) | 71,237 | |
| | ASVspoof2021_DF_raw (Yamagishi et al., 2021) | 611,829 | |
| | ASVspoof2021_LA_raw (Yamagishi et al., 2021) | 181,566 | |
| | ASVspoof2021_PA_raw (Yamagishi et al., 2021) | 943,110 | Replayed real speech labeled as real |
| | ASVspoof5_raw (Wang et al., 2024) | 680,774 | Vocoder generated samples labeled as fake |
| | AugmentedDeepfake-2k | $2.16 \times 10^6$ | Self-curated dataset with 10 augs applied per file |
| | CFAD_raw (Ma et al., 2024) | 59,500 | Partial fake samples excluded |
| | Codecfake_EN_CH_raw (Wu et al., 2024) | 322,015 | Codec generated samples labeled as fake |
| | CodecFake_EN_raw (Xie et al., 2025a) | 707,872 | Codec generated samples labeled as fake |
| | CVoiceFake_raw (Li et al., 2024) | 1,466,578 | Vocoder generated samples labeled as fake |
| | DECRO_raw (Ba et al., 2023) | 37,314 | |
| | Deepfake_eval_2024_preprocessed (Chandra et al., 2025) | 66,282 | |
| | DFADD_raw (Du et al., 2024) | 3,755 | |
| | DiffSSD_raw (Bhagtani et al., 2025) | 54,613 | |
| | DiffuseOrConfuse_raw (Firc et al., 2024) | 196,500 | |
| | DSD_Corpus_preprocessed (Doan et al., 2024) | 255,655 | |
| | ELTOLSM_preprocessed (Demirörs et al., 2025) | 10,849 | |
| | EmoSpoofTTS_preprocessed (Mahapatra et al., 2025) | 49,936 | |
| | FoR-2seconds_raw (Reimao & Tzerpos, 2019) | 1,088 | |
| | FoR-norm_raw (Reimao & Tzerpos, 2019) | 4,634 | |
| | FoR-rerecorded_raw (Reimao & Tzerpos, 2019) | 816 | |
| | HABLA_raw (Flórez et al., 2023) | 32,327 | |
| | IndieFake_Dataset_preprocessed (Kumar et al., 2025) | 22,214 | |
| | InTheWild_raw (Müller et al., 2022) | 31,779 | |
| | JMD_Corpus_preprocessed (Takamichi, 2021a) | 7,094 | |
| | Jspaw_ver2_LA_preprocessed (Takamichi, 2024) | 1,244 | |
| | Jspaw_ver2_PA_preprocessed (Takamichi, 2024) | 91,918 | |
| | JSSS_Corpus_preprocessed (Takamichi et al., 2020a) | 14,142 | |
| | JSUT_preprocessed (Sonobe et al., 2017) | 14,205 | |

*Continued on next page...*

*Table 15.* Evaluation datasets categorized by tasks (Continued)

| Task | Dataset | #Samples | Notes |
|---|---|---|---|
| | JVS_Corpus_preprocessed (Takamichi et al., 2020b) | 46,483 | |
| | KoreanReadSpeechCorpus_preprocessed (Deeply Inc., 2021) | 4,700 | |
| | KSS_preprocessed (Park, 2019; Park & Mulc, 2019) | 8,772 | Vocoder generated samples labeled as fake |
| | LibreSeVoc_raw (Sun et al., 2023) | 92,406 | |
| | MLAAD_v3_en_preprocessed (Müller et al., 2024) | 7,129 | Partial overlap with pretraining data |
| | MLAAD_v3_v4_preprocessed (Müller et al., 2024) | 905,909 | Partial overlap with pretraining data |
| | MLAAD_v6_preprocessed (Müller et al., 2024) | 439,573 | |
| | MSceneSpeech_raw (Yang et al., 2024) | 1,004 | |
| | ODSS_raw (Yaroshchuk et al., 2023) | 30,025 | |
| | ReMASC_raw (Gong et al., 2019) | 17,582 | Replayed real speech labeled as real |
| | ReplayDF_preprocessed (Müller et al., 2025) | 140,661 | Replayed real speech labeled as real |
| | RFP_raw (AlAli & Theodorakopoulos, 2023) | 119,921 | Partial fake samples excluded |
| | Seoul_Corpus_preprocessed (Bang et al., 2020) | 78,515 | |
| | ShiftySpeech_raw (Garg et al., 2025) | 2,781,458 | Partial overlap with pretraining data |
| | SpeechFake_MD_v1_preprocessed (Huang et al., 2025) | 819,275 | |
| | SpoofCeleb_preprocessed (Jung et al., 2025a) | 81,720 | |
| | SRC4VC_raw (Saito et al., 2024) | 10,400 | |
| | STCodecfake_raw (Xie et al., 2025b) | 319,014 | Codec generated samples labeled as fake |
| | TIMIT-TTS_raw (Salvi et al., 2023) | 79,120 | |
| | Tri_jek_v1_preprocessed (Takamichi, 2021b) | 17,271 | |
| **SVDD** | CtrSVDD-raw (Zang et al., 2024) | 92,769 | |
| **PFSL** | PartialSpoof-raw (Zhang et al., 2022) | 121,461 | |
| | Half-Truth-raw (Yi et al., 2021) | 107,224 | |
| | LlamaPartialSpoof-raw (Luong et al., 2025) | 42,767 | |
| **ST** | ASVspoof5-ST-test (Wang et al., 2024) | 6,808 | Test split of ASVspoof5-ST |
| **AVDD** | FakeAVCeleb-raw (Khalid et al., 2021) | 65,606 | We created 3s video segments for evaluation |
| | PolyGlotFake-raw (Hou et al., 2024) | 98,411 | |

# G. Other Experimental Results

## G.1. SDD Model Comparison

Table 16 and 17 provide per-dataset EER and accuracy of Alethia-Large and W1V-1B along with their performance difference under the EXPANDED-AUG and EXPANDED condition, respectively. For both conditions, Alethia-large shows better performance with significantly fewer datasets with accuracies below 90% and 80%. Figure. 5 and Figure. 6 further visualize the performance difference between Alethia-Large and W2V-1B under the EXPANDED-AUG condition and EXPANDED condition, respectively. Under the EXPANDED-AUG condition, among the 50 SDD datasets we observe consistently better performance on 40 datasets with Alethia-Large in both EER and accuracy, with up to 17.5% increase in accuracy and 3.6% decrease in EER. A similar trend is seen with the EXPANDED condition, where Alethia-Large consistently outperforms W2V-1B on 36 out of 50 datasets, with up to 16.1% accuracy gain and 5.6% decrease in EER.

*Table 16.* SDD per-dataset performance comparison between Alethia-Large and W2V-1B under the EXPANDED-AUG condition. Datasets with just one class have EER values of −1. Accuracies between 80% to 90% are marked in red, below 80% are in orange.

| Dataset | Alethia-Large | | W2V-1B | | | |
|---|---|---|---|---|---|---|
| | **Accuracy** | **EER** | **Accuracy** | **EER** | **Accuracy Diff** | **EER Diff** |
| ALM-ADD_raw | 87.68 | 0.1244 | 81.79 | 0.1410 | 5.89 | −0.0166 |
| ASVspoof2019_LA_raw | 98.59 | 0.0118 | 98.85 | 0.0076 | −0.26 | 0.0042 |
| ASVspoof2021_DF_raw | 95.05 | 0.0314 | 97.65 | 0.0258 | −2.60 | 0.0056 |
| ASVspoof2021_LA_raw | 93.64 | 0.0525 | 93.37 | 0.0577 | 0.27 | −0.0052 |
| ASVspoof2021_PA_raw | 95.73 | −1 | 97.48 | −1 | −1.75 | - |
| ASVspoof5_raw | 82.52 | 0.1178 | 82.71 | 0.1138 | −0.19 | 0.004 |
| AugmentedDeepfake-2k_preprocessed | 99.19 | 0.0039 | 98.52 | 0.0059 | 0.67 | −0.002 |
| CFAD_raw | 98.92 | 0.0121 | 97.17 | 0.0233 | 1.75 | −0.0112 |
| Codecfake_EN_CH_raw | 59.36 | 0.1656 | 56.82 | 0.1934 | 2.54 | −0.0278 |

| Dataset | Alethia-Large | | W2V-1B | | | |
|---|---|---|---|---|---|---|
| | **Accuracy** | **EER** | **Accuracy** | **EER** | **Accuracy Diff** | **EER Diff** |
| CodecFake_EN_raw | 60.08 | 0.1302 | 57.66 | 0.1421 | 2.42 | −0.0119 |
| CVoiceFake_raw | 90.66 | 0.0355 | 90.74 | 0.0285 | −0.08 | 0.0070 |
| DECRO_raw | 93.34 | 0.0509 | 91.75 | 0.0468 | 1.59 | 0.0041 |
| Deepfake_eval_2024_preprocessed | 91.12 | 0.1162 | 89.75 | 0.1291 | 1.37 | −0.0129 |
| DFADD_raw | 97.79 | 0.0043 | 94.01 | 0.0087 | 3.78 | −0.0044 |
| DiffSSD_raw | 99.85 | 0.0009 | 99.49 | 0.0020 | 0.36 | −0.0011 |
| DiffuseOrConfuse_raw | 98.76 | 0.0044 | 96.86 | 0.0081 | 1.90 | −0.0037 |
| DSD_Corpus_preprocessed | 95.21 | 0.0665 | 94.44 | 0.0614 | 0.77 | 0.0051 |
| ELTOLSM_preprocessed | 96.69 | −1 | 92.33 | −1 | 4.36 | - |
| EmoSpoofTTS_preprocessed | 97.66 | 0.0193 | 91.58 | 0.051 | 6.08 | −0.0317 |
| FoR-2seconds_raw | 99.82 | 0 | 100 | 0 | −0.18 | 0 |
| FoR-norm_raw | 99.46 | 0.0022 | 98.81 | 0.0034 | 0.65 | −0.0012 |
| FoR-rerecorded_raw | 90.69 | 0.0711 | 85.17 | 0.1078 | 5.52 | −0.0367 |
| HABLA_raw | 99.83 | 0.0015 | 99.87 | 0.0011 | −0.04 | 0.0004 |
| IndieFake_Dataset_preprocessed | 94.96 | 0.0473 | 94.63 | 0.0476 | 0.33 | −0.0003 |
| InTheWild_raw | 98.79 | 0.012 | 98.69 | 0.0135 | 0.10 | −0.0015 |
| JMD_Corpus_preprocessed | 90.8 | −1 | 93.08 | −1 | −2.28 | - |
| Jspaw_ver2_LA_preprocessed | 98.15 | −1 | 98.39 | −1 | −0.24 | - |
| Jspaw_ver2_PA_preprocessed | 87.32 | −1 | 86.20 | −1 | 1.12 | - |
| JSSS_Corpus_preprocessed | 91.91 | −1 | 88.44 | −1 | 3.47 | - |
| JSUT_preprocessed | 99.8 | −1 | 99.81 | −1 | −0.01 | - |
| JVS_Corpus_preprocessed | 65.98 | −1 | 48.64 | −1 | 17.34 | - |
| KoreanReadSpeechCorpus_preprocessed | 96.98 | −1 | 98.96 | −1 | −1.98 | - |
| KSS_preprocessed | 99.59 | −1 | 99.42 | −1 | 0.17 | - |
| LDC_MLCTS_v1.0.0_preprocessed | 97.21 | −1 | 94.79 | −1 | 2.42 | - |
| LibreSeVoc_raw | 99.94 | 0.0012 | 99.95 | 0.0007 | −0.01 | 0.0005 |
| MLAAD_v3_en_preprocessed | 95.01 | −1 | 93.58 | −1 | 1.43 | - |
| MLAAD_v3_v4_preprocessed | 97.39 | 0.0415 | 96.17 | 0.0564 | 1.22 | −0.0149 |
| MLAAD_v6_preprocessed | 92.27 | −1 | 86.86 | −1 | 5.41 | - |
| MSceneSpeech_raw | 95.32 | −1 | 94.52 | −1 | 0.80 | - |
| ODSS_raw | 95.05 | 0.0266 | 89.17 | 0.0432 | 5.88 | −0.0166 |
| ReMASC_raw | 77.71 | 0.1714 | 75.35 | 0.1908 | 2.36 | −0.0194 |
| ReplayDF_preprocessed | 82.88 | 0.1147 | 77.25 | 0.1369 | 5.63 | −0.0222 |
| RFP_raw | 94.20 | 0.0455 | 93.71 | 0.0418 | 0.49 | 0.0037 |
| Seoul_Corpus_preprocessed | 99.61 | −1 | 98.41 | −1 | 1.20 | - |
| ShiftySpeech_raw | 88.45 | 0.0816 | 86.98 | 0.0859 | 1.47 | −0.0043 |
| SpeechFake_MD_v1_preprocessed | 97.20 | −1 | 93.74 | −1 | 3.46 | - |
| SpoofCeleb_preprocessed | 98.51 | 0.0430 | 98.04 | 0.0509 | 0.47 | −0.0079 |
| SRC4VC_raw | 99.27 | 0.0067 | 98.59 | 0.0123 | 0.68 | −0.0056 |
| STCodecfake_raw | 81.69 | 0.0703 | 77.81 | 0.0767 | 3.88 | −0.0064 |
| TIMIT-TTS_raw | 91.59 | −1 | 89.27 | −1 | 2.32 | - |
| Tri_jek_v1_preprocessed | 97.39 | −1 | 96.43 | −1 | 0.96 | - |

*Table 17.* SDD per-dataset performance comparison between Alethia-Large and W2V-1B under the EXPANDED condition. Datasets with just one class have EER values of −1. Accuracies between 80% to 90% are marked in red, below 80% are in orange.

| Dataset | Alethia-Large | | W2V-1B | | | |
|---|---|---|---|---|---|---|
| | **Accuracy** | **EER** | **Accuracy** | **EER** | **Accuracy Diff** | **EER Diff** |
| ALM-ADD_raw | 92.44 | 0.1090 | 85.71 | 0.1244 | 6.73 | −0.0154 |
| ASVspoof2019_LA_raw | 98.89 | 0.0110 | 97.37 | 0.0126 | 1.52 | −0.0016 |
| ASVspoof2021_DF_raw | 94.99 | 0.0436 | 96.36 | 0.0278 | −1.37 | 0.0158 |
| ASVspoof2021_LA_raw | 95.35 | 0.0516 | 90.68 | 0.0643 | 4.67 | −0.0127 |
| ASVspoof2021_PA_raw | 96.64 | −1 | 98.88 | −1 | −2.24 | - |
| ASVspoof5_raw | 91.95 | 0.0944 | 83.2 | 0.0887 | 8.75 | 0.0057 |
| AugmentedDeepfake-2k_preprocessed | 95.16 | 0.0624 | 90.61 | 0.1142 | 4.55 | −0.0518 |
| CFAD_raw | 87.9 | 0.0966 | 80.76 | 0.1109 | 7.14 | −0.0143 |
| Codecfake_EN_CH_raw | 75.12 | 0.1004 | 59.88 | 0.1412 | 15.24 | −0.0408 |

| Dataset | Alethia-Large | | W2V-1B | | Accuracy Diff | EER Diff |
|---|---|---|---|---|---|---|
| | Accuracy | EER | Accuracy | EER | | |
| CodecFake_EN_raw | 75.6 | 0.1056 | 62.09 | 0.1307 | 13.51 | −0.0251 |
| CVoiceFake_raw | 62.31 | 0.1175 | 54.22 | 0.1151 | 8.09 | 0.0024 |
| DECRO_raw | 93.76 | 0.0476 | 88.8 | 0.0516 | 4.96 | −0.0040 |
| Deepfake_eval_2024_preprocessed | 89.41 | 0.1303 | 86.98 | 0.1648 | 2.43 | −0.0345 |
| DFADD_raw | 99.36 | 0.0050 | 93.16 | 0.0146 | 6.20 | −0.0096 |
| DiffSSD_raw | 88.34 | 0.2043 | 89.77 | 0.2275 | −1.43 | −0.0232 |
| DiffuseOrConfuse_raw | 93.64 | 0.1498 | 94.66 | 0.2065 | −1.02 | −0.0567 |
| DSD_Corpus_preprocessed | 96.52 | 0.0455 | 94.15 | 0.0559 | 2.37 | −0.0104 |
| ELTOLSM_preprocessed | 98.17 | −1 | 94.95 | −1 | 3.22 | - |
| EmoSpoofTTS_preprocessed | 96.52 | 0.0334 | 90.94 | 0.0599 | 5.58 | −0.0265 |
| FoR-2seconds_raw | 98.44 | 0.0018 | 99.54 | 0.0055 | −1.10 | −0.0037 |
| FoR-norm_raw | 97.84 | 0.0066 | 98.36 | 0.008 | −0.52 | −0.0014 |
| FoR-rerecorded_raw | 87.38 | 0.1324 | 84.56 | 0.1103 | 2.82 | 0.0221 |
| HABLA_raw | 99.26 | 0.0036 | 99.83 | 0.0013 | −0.57 | 0.0023 |
| IndieFake_Dataset_preprocessed | 95.56 | 0.0437 | 95.19 | 0.042 | 0.37 | 0.0017 |
| InTheWild_raw | 99.07 | 0.0109 | 98.95 | 0.0111 | 0.12 | −0.0002 |
| JMD_Corpus_preprocessed | 95.14 | −1 | 93.78 | −1 | 1.36 | - |
| Jspaw_ver2_LA_preprocessed | 100 | −1 | 99.28 | −1 | 0.72 | - |
| Jspaw_ver2_PA_preprocessed | 97.55 | −1 | 99.33 | −1 | −1.78 | - |
| JSSS_Corpus_preprocessed | 97.86 | −1 | 82.96 | −1 | 14.90 | - |
| JSUT_preprocessed | 99.98 | −1 | 99.92 | −1 | 0.06 | - |
| JVS_Corpus_preprocessed | 74.63 | −1 | 58.52 | −1 | 16.11 | - |
| KoreanReadSpeechCorpus_preprocessed | 99.68 | −1 | 99.85 | −1 | −0.17 | - |
| KSS_preprocessed | 98.58 | −1 | 98.89 | −1 | −0.31 | - |
| LDC_MLCTS_v1.0.0_preprocessed | 99.81 | −1 | 99.44 | −1 | 0.37 | - |
| LibreSeVoc_raw | 98.2 | 0.0082 | 94.63 | 0.0085 | 3.57 | −0.0003 |
| MLAAD_v3_en_preprocessed | 95.12 | −1 | 92.03 | −1 | 3.09 | - |
| MLAAD_v3_v4_preprocessed | 97.6 | 0.0356 | 96.35 | 0.0561 | 1.25 | −0.0205 |
| MLAAD_v6_preprocessed | 92.16 | −1 | 84.9 | −1 | 7.26 | - |
| MSceneSpeech_raw | 89.24 | −1 | 98.11 | −1 | −8.87 | - |
| ODSS_raw | 97.43 | 0.0217 | 88.94 | 0.0482 | 8.49 | −0.0265 |
| ReMASC_raw | 70.05 | 0.2080 | 64.81 | 0.2346 | 5.24 | −0.0266 |
| ReplayDF_preprocessed | 81.55 | 0.1262 | 73.63 | 0.1422 | 7.92 | −0.016 |
| RFP_raw | 92.42 | 0.0648 | 91.34 | 0.0735 | 1.08 | −0.0087 |
| Seoul_Corpus_preprocessed | 99.03 | −1 | 99.23 | −1 | −0.20 | - |
| ShiftySpeech_raw | 78.37 | 0.1222 | 71.07 | 0.137 | 7.30 | −0.0148 |
| SpeechFake_MD_v1_preprocessed | 95.5 | −1 | 91.33 | −1 | 4.17 | - |
| SpoofCeleb_preprocessed | 98.12 | 0.0528 | 98.57 | 0.0378 | −0.45 | 0.0150 |
| SRC4VC_raw | 99.21 | 0.0073 | 97.72 | 0.0142 | 1.49 | −0.0069 |
| STCodecfake_raw | 85.32 | 0.0644 | 76.86 | 0.0902 | 8.46 | −0.0258 |
| TIMIT-TTS_raw | 94.85 | −1 | 91.61 | −1 | 3.24 | - |
| Tri_jek_v1_preprocessed | 85.71 | −1 | 89.62 | −1 | −3.91 | - |

### G.2. Embedding Visualization

Figure. 7(a). shows UMAP visualizations of pretrained encoder embedding space on the test split of ASVspoof5-ST dataset. Audio is resampled to 16 kHz and duration-normalized to 4s via random cropping or circular padding. For each pretrained encoder, we extract the final-layer representations and apply average pooling over time axis to obtain one fixed-dimensional embedding per utterance. The 2D UMAP projections are computed using cosine distance ($n_{neighbors} = 15$, $min\_dist = 0.1$). All silhouette scores shown in the figure are computed on the pooled final-layer embeddings. Across encoders, Alethia-Base yields the most structured representations for the source labels, and is the only model with a positive silhouette score. Similarly, Figure. 7(b) visualizes the resultant 17-D classifier outputs from the frozen encoder+MLP finetuned models, with the same UMAP settings. Again, Alethia-Base as the frozen encoder produces clearly separated clusters with a much higher silhouette score ($S = 0.47$), compared to weaker separation for WavLM-Large ($S = 0.22$) and limited separation for HuBERT-Large and W2V-XLSR-300M. These results suggest Alethia-Base provides a more linearly separable embedding geometry for the underlying attack categories, benefiting lightweight downstream classification.

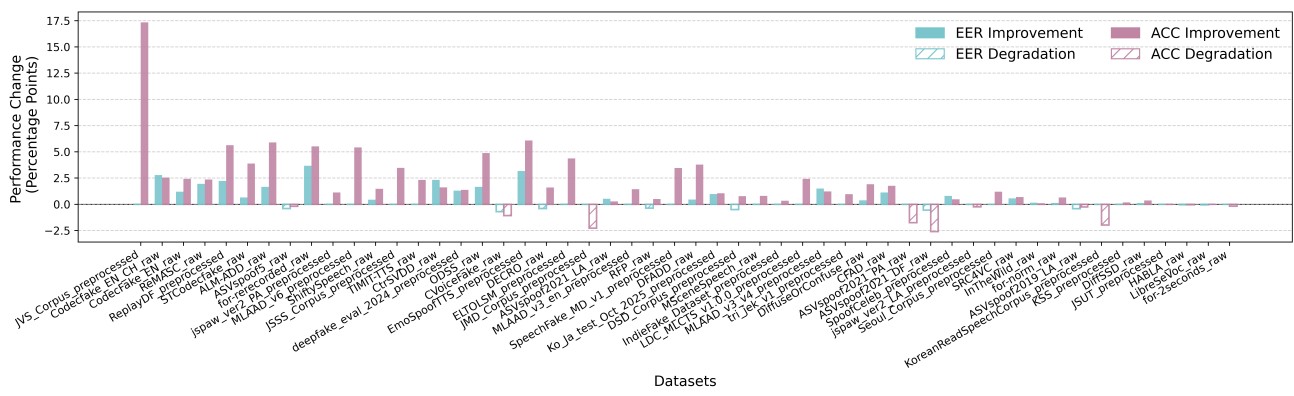

*Figure 5.* Comparison between Alethia-Large and W2V-1B under the EXPANDED-AUG condition. Positive values suggest performance improvement.

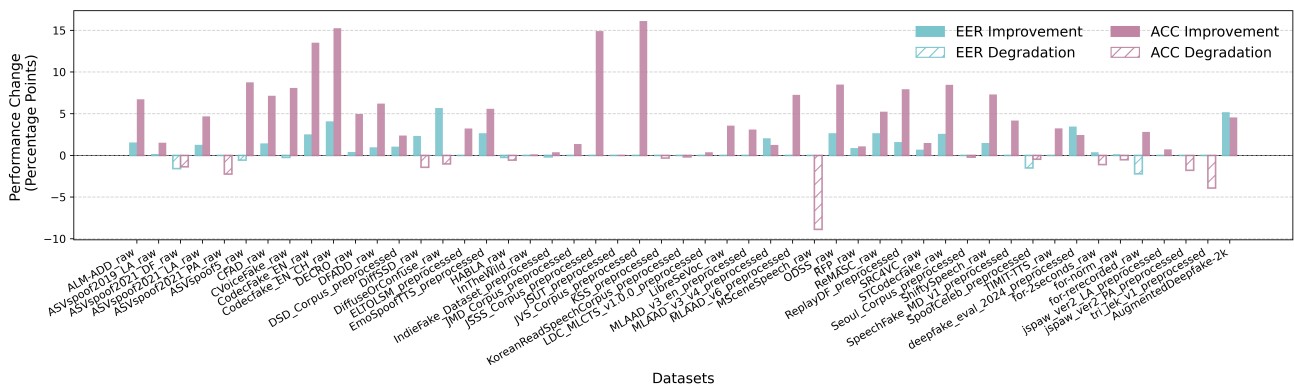

*Figure 6.* Comparison between Alethia-Large and W2V-1B under the EXPANDED condition. Positive values suggest performance improvement.

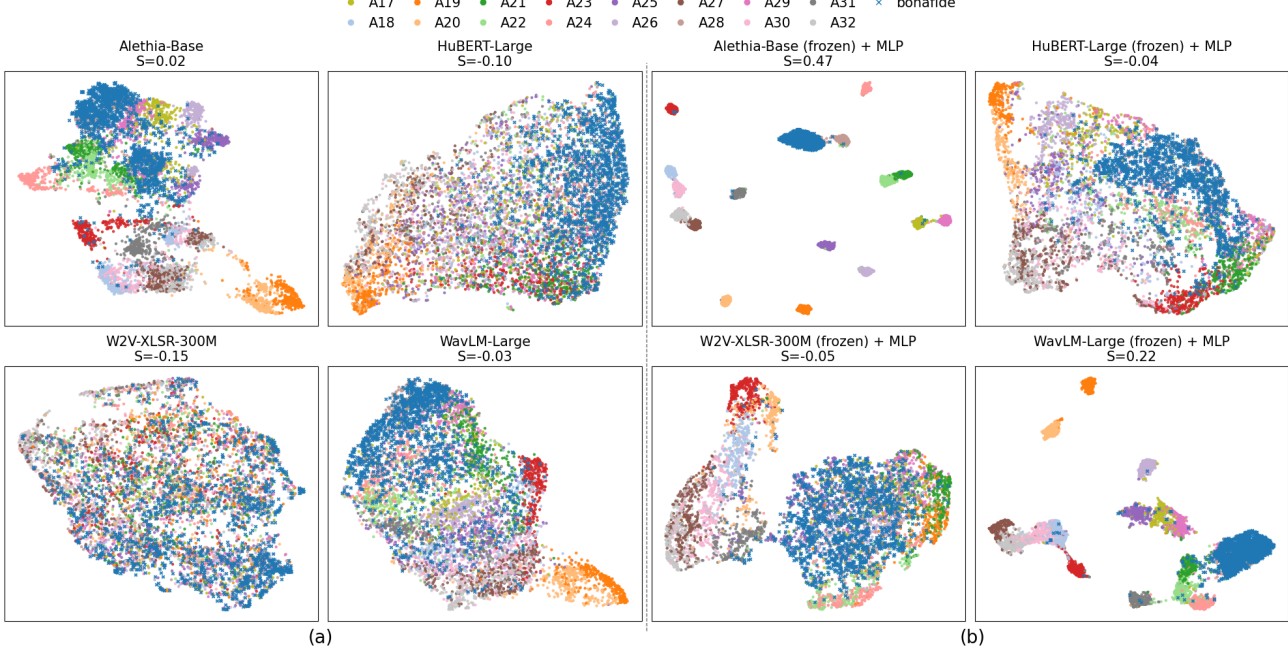

*Figure 7.* For ASVspoof5-ST test split: a) UMAP visualizations of pretrained embeddings b) final layer embeddings from finetuned frozen-backbone + MLP setup for ST. Points are colored by attack label, and $S$ denotes the Silhouette Score.

*Table 18.* Model performance on AVDD task in the zero-shot setting comparing additional metrics.

| Model | EER ↓ | Acc ↑ | TPR ↑ | TNR ↑ |
|---|---|---|---|---|
| *FakeAVCeleb dataset* | | | | |
| HuBERT-Large | 8.1 | **93.3** | 97.9 | 89.5 |
| WavLM-Large | 7.0 | 92.8 | 95.6 | 90.5 |
| W2V-XLSR-300M | **5.8** | 92.4 | **99.6** | 86.3 |
| Alethia-Base | 6.3 | 92.1 | **99.6** | 85.7 |
| *PolyGlotFake dataset* | | | | |
| HuBERT-Large | 13.9 | 71.8 | 70.8 | 92.7 |
| WavLM-Large | 14.1 | 67.0 | 65.6 | **93.9** |
| W2V-XLSR-300M | 9.4 | **94.3** | **94.7** | 87.0 |
| Alethia-Base | **7.1** | 94.2 | 94.4 | 91.3 |

## G.3. AVDD Results

Table 18 presents additional metrics (accuracy, TPR, and TNR) for comparing Alethia with the baseline models to supplement Table 8. While WavLM-Large and HuBERT-Large have good performance on the FakeAVCeleb dataset, we observe a sharp drop in TPR on the PolyGlotFake dataset, which includes more challenging deepfake samples. Alethia-Base has the lowest EER on PolyGlotFake and consistently high values for the other threshold-based metrics. These results indicate the strong potential for our proposed foundation model and pretraining recipes to lead to improvements in the multi-modal setting where the detectors can leverage complimentary deepfake cues.

