# OpenReview forum: "Alethia: a Foundational Encoder for Voice Deepfakes"
_ICML.cc/2026/Conference — ICML 2026 regular_

### Official Review · Reviewer_cnqW · 2026-03-09

**Soundness:** 3
**Presentation:** 3
**Significance:** 3
**Originality:** 2
**Overall Recommendation:** 4
**Confidence:** 3

**Summary:**

This paper proposes Alethia, a foundational audio encoder tailored for voice deepfake detection and related tasks. The key idea is a dual-branch self-supervised pretraining recipe that couples bottleneck masked embedding prediction of continuous, multi-layer teacher features with a flow-matching spectrogram reconstruction objective. The model was pre-trained on 19,000 hours of real and fake speech and comprehensively evaluated on 56 datasets across 5 downstream tasks.

**Compliance With Llm Reviewing Policy:**

Affirmed.

**Final Justification:**

After the rebuttal process, my concerns are addressed. I incline to give weak accept to this paper.

**Key Questions For Authors:**

1.Which specific TTS/VC models were used to synthesize the 18k-hour set? How many overlap with the evaluation corpora? Can you provide the exhaustive list of the 20 generative models?
2.Can you provide a standard continuous-embedding distillation baseline pretrained on the exact same 19k-hour dataset?
3.What is the observed effect of Hungarian coupling vs. simple Gaussian pairing on convergence, reconstruction quality at masked positions, and downstream metrics? Any ablation on σ_min and σ_eps?

**Limitations:**

yes

**Strengths And Weaknesses:**

1.The paper evaluates the results on 5 different tasks (SDD, SVDD, PFSL, ST, AVDD) and 56 datasets. This large-scale cross-validation enhances the statistical reliability of the conclusions.However, Alethia is pretrained on 19k hours of curated, deepfake-heavy audio, whereas the baseline general-purpose SFMs lack exposure to this specific distribution. This discrepancy makes it ambiguous whether the observed performance gains stem from the proposed dual-branch architectural innovations or simply from the domain-specific data advantage.
2.The paper proposed to mitigate the discriminative power degradation caused by joint prediction and generation pre-training. By aligning the multi-layered manifold of the teacher model with Bottleneck MEP and supplementing it with spectrogram reconstruction based on Conditional Flow-matching, information about generation defects is effectively preserved.
3.The motivation is clearly laid out, including early negative results with masked-token pretraining and the Mutual information study. The use of per-layer 2D masking (time and channels) and all-steps supervision for continuous embedding prediction is well-motivated and empirically justified via stability and convergence analysis.
4.The paper claims this is the first "base encoder" for speech deepfakes. However, from a technical implementation perspective, Alethia's pre-training heavily relies on frozen teacher models to extract target representations. This is essentially a task-aware distillation, rather than a true "base model" that learns the data distribution from scratch.
5.The paper used 20 different off-the-shelf generative models to construct 18,000 hours of synthetic data. However, neither the main text nor the appendix clearly explains the specific information about these models. This makes it difficult to assess and verify whether these models overlap with the attack methods in the evaluation test set, weakens the paper's claims about performance improvements.

---

> ### Author Rebuttal · Authors · 2026-03-30
>
> Thank you for your valuable feedback. Please find our responses below.
>
> > 1. *"However, Alethia is pretrained . . . domain-specific data advantage."*
>
> We agree that isolating data effects is critical. In Section 3, we conducted controlled experiments with the **100M-parameter** Base versions of Wav2vec2 and HuBERT models. We kept the original pretraining recipes and exposed the models to 1k hours of real and deepfake speech. As shown in Table 1, **standard off-the-shelf pretraining recipes yielded performance degradations (positive $\Delta$EER) in three scenarios and a negligible gain in the fourth scenario**. These observations confirm that simply exposing an SFM model to a deepfake data distribution is insufficient. For a direct comparison, a **100M version of Alethia pretrained on the same data showed significant performance gains over these baselines (EER reduced by 3.05%)**. This shows the strength of the proposed dual-branch recipe in extracting useful discriminative features. We will update Table 1 with the 100M Alethia results in the revised version to further clarify this distinction.
>
> > 2. *"The paper claims this is the first . . . data distribution from scratch."*
>
> We appreciate the reviewer’s perspective on the definition of foundation models and the distillation training paradigm. We characterized Alethia as a foundational encoder rather than a task-distilled model due to fundamental differences from previous speech encoders and the impact we note on various deepfake detection tasks.
>
> - **The use of teacher models in historical foundation models:** The inclusion of teacher models is a standard and successful practice in developing state-of-the-art speech foundation models. For example, WavLM utilizes quantized frozen HuBERT embeddings as targets (https://arxiv.org/abs/2110.13900).
>
> - **Fundamentally Different Pretraining Scheme:** Alethia replaces standard discrete, quantized targets, which from our mutual information analysis proved to be uninformative for deepfake traces (Section 3), with continuous bottleneck masked embedding prediction to prevent information loss.
>
> - **Generative Inductive Bias:** We incorporate a parallel flow-matching branch for spectrogram reconstruction, providing an inductive bias essential for capturing sub-perceptual artifacts from modern deepfake synthesizers.
>
> - **Zero-shot generalizability to unseen data distributions:** Alethia demonstrates strong zero-shot performance on domains entirely unseen by the teacher or in the pretraining data. For instance, Alethia-Large achieves a 10.8% EER on singing voice deepfakes (SVDD), **outperforming in-domain baselines specifically optimized for that task** (Table 5).
>
> > 3. *"The paper used 20 different . . . claims about performance improvements. . . Which specific TTS/VC . . . 20 generative models?"*
>
> - **Model List:** The 20 generative models include a diverse range of SOTA TTS and VC frameworks such as Commercial platforms (e.g., ElevenLabs), VITS, YourTTS, TorToiSe, and various diffusion-based synthesizers. We will provide the exhaustive list in the revised Appendix.
> - **Overlap and Integrity:** While a strictly zero model overlap is difficult/cumbersome to achieve when running evaluations on a large scale, we have explicitly avoided using code or hyper-parameters provided in any of the evaluation repositories. By focusing on evaluations at scale (56 datasets), our goal is to assess whether Alethia learns generalizable generative artifacts rather than overfitting to certain dataset-specific model signatures. Among the 56 evaluation datasets, those that have explicit overlap with pretraining data, albeit without ground-truth labels, have been duly demarcated in Table 15.
>
> > 4. *“Can you provide a standard . . . same 19k-hour dataset?”*
>
> Given the time constraints of the rebuttal phase, we provide the following technical justification and ablation results to demonstrate why Alethia’s architecture is a superior alternative to standard distillation:
>
> - **Inefficiency of Standard Layer-to-Layer Distillation:** Standard layer-to-layer distillation baseline is often used for compressing large foundation models (https://ieeexplore.ieee.org/document/9747490), and can be suboptimal for pretraining a stronger student model. This is because standard distillation often results in a student model strictly bounded by the teacher’s performance. We therefore propose the bottleneck architecture which is shown to surpass the teacher’s performance.
> - **Bottleneck as a Powerful Prior:** Even when masking and generative objectives are removed, the bottleneck architecture alone achieves a 0.7% EER improvement over a standard WavLM-Large baseline on the same data (Section 6.6). This demonstrates that our architectural choices serve as a superior prior for the deepfake manifold compared to standard distillation.
>
> ### Additional loss ablations
> We request the reviewer to refer to our response to **reviewer A4Sm, point-1** on a related comment.

---

> > ### Author Rebuttal · Reviewer_cnqW · 2026-04-02
> >
> > I incline to raise my rate score.

---

### Official Review · Reviewer_2YEr · 2026-03-10

**Soundness:** 4
**Presentation:** 4
**Significance:** 3
**Originality:** 3
**Overall Recommendation:** 5
**Confidence:** 3

**Summary:**

The paper introduces Alethia, a novel foundational audio encoder designed for detecting deepfakes. The authors argue that existing speech foundational models (SFMs) such as HuBERT are limited because their pretraining relies on discrete masked tokens predictions. The quantization process naturally dicards information that may becritical for detecting deepfakes.

In order to solve this issue, Alethia employs a dual-branch training recipe: the first branch performs masked embeddings prediction (while distilling knowledge from a frozen teacher model), the second branch reconstructs the unmasked spectrogram using conditional flow matching. The second branch forces the model to learn low-level acoustic signal that can be essential for spotting deepfakes.

Alethia is trained on 19k hours of speech which includes curated deepfakes. Alethia was evaluated actoss 56 datasets spanning 5 tasks. The results showcase that it outperforms existing SFMs.

Contributions:
* speech foundational model specifically geared towards deepfake detection
* Dual-branch pretraining recipe: using a first branch that learns embeddings and a second branch forcing the model to learn acoustic signal by attempting to reconstruct the masked spectrogram.
* Zero-shot generalizability: Alethia performed well against unseen attacks such as singing deepfakes.

**Compliance With Llm Reviewing Policy:**

Affirmed.

**Final Justification:**

I increased my score from a 'Weak accept' to 'Accept'. I think the authors addressed my questions. However, I feel this was a borderline decision as I like the conceptual idea but I am skeptical it can be easily adopted by the community when the code is not available.

**Key Questions For Authors:**

* Reproducibility: Do you plan on releasing the model weights or code as to make the claims more verifiable?I am willing to raise my score if this is addressed.
* Initialization strategy: given that the 1B and 400M parameter models were trained for only a single epoch, were the student encoders initialized with random weights or from existing pre-trained checkpoints (e.g., WavLM or Wav2vec2)?
* Loss Calculation on Unmasked Frames: you note that calculating the prediction loss over all time steps stabilizes training. With masking ratios of 10%, how do you ensure the model does not simply learn a trivial identity mapping because of the 90% unmasked audio?
* Loss Balancing: Can you provide a scale analysis of the total pretraining loss to confirm that one loss objective does not trivially dominate the gradients throughout the training steps?

**Limitations:**

yes

**Strengths And Weaknesses:**

Strengths:
* Commendable evaluation scale: the benchmarking is really rigorous. Using 56 datasets across 5 tasks. the evaluation and ablation studies were comprehensive.
* The methodlogy is well-motivated: the authors make a clear assumption that using discrete quantization targets in SFMs is limited theoretically. Also, the dual-branch design is well motivated as the intuition that reconstructing the spectrogram to force the model into learning acoustic information makes sense, and is substantiated by experiments.
* The zero-shot performance is promising.
* Clear ablation studies: The paper effectively isolates its design choices, successfully proving that the generative flow-matching branch is strictly necessary for detecting high-fidelity diffusion and flow-matching deepfakes.

Weaknesses:
* Not reproducible: I may be mistaken, but the paper doesn't promise to release any of the following:dataset, code, model weights. I can understand that some of these can be proprietary, but open-sourcing none of the above makes the claims in the paper unverifiable.
* Unfair comparisons: Alethia was trained with deepfakes in mind. HuBERT,for example, wasn't trained with deepfake detection in mind. The authors training the HuBERT model with added deepfakes data is a good step towards determining the effect of the added deepfake data, but I don't know if the amount of deepfakes in both trainings (Alethia vs HuBERT) is the same. It is difficult to isolate if Alethia is simply outperforming the baseline because of its exposure to deepfakes.
* Reliance on frozen teacher: the student model is forced to predict the continuous embeddings of the frozen teacher. Potentially, the student may not be able to learn deepfake patterns beyond what the frozen teacher model allows for.
* The prediction loss logic is not well-motivated: The authors calculate the continuous embedding prediction loss over all time steps, rather than exclusively at masked positions. Because the masking ratio is only about 10%, 90% of the signal is unmasked. This heavily risks the encoder collapsing into a trivial identity mapping for the vast majority of the sequence.

---

> ### Author Rebuttal · Authors · 2026-03-31
>
> Thank you for the valuable feedback.
>
> 1. *Reproducibility*
>
> Regarding the reproduction of our results and the availability of our model:
> - **Technical Transparency**: While we are not releasing pretrained checkpoints at this time, we have provided exhaustive architectural details. Appendix B and Table 10 specify all hyperparameters, including masking, pretraining, and finetuning details. We made a sincere effort to ensure sufficient detail for reproducibility.
> - **Data**: We have documented our quality control pipeline (Appendix E.1 - Preprocessing, Table 12), including VAD, multi-speaker rejection, intelligibility estimation, and duration control, to ensure our 19k-hour pretraining corpus can be replicated. We also provide a detailed list of datasets used in this study along with the references to facilitate reproducible results (Table 15).
> - **Future Release**: We are discussing an open-source release of the code and weights, pending institutional review of various aspects.
>
> 2. *Unfair comparisons*
>
> We agree that isolating data effects is critical., In Section 3, we conducted controlled experiments with the 100M-parameter **Base** versions of Wav2vec2 and HuBERT models. We kept the original pretraining recipes and exposed the models to 1k hours of real and deepfake speech. As shown in Table 1, standard off-the-shelf pretraining recipes yielded performance degradations (positive $\Delta$EER) in three scenarios and a negligible gain in the fourth scenario. These observations confirm that simply exposing an SFM to a deepfake data distribution is insufficient. For a direct comparison, a 100M version of Alethia pretrained on the **same data** (1k hours) showed significant performance gains over these baselines (EER reduced by 3.05%). This shows the strength of the proposed dual-branch pretraining recipe in extracting useful discriminative features. We will update Table 1 with the 100M Alethia results in the revised version to further clarify this distinction.
>
> 3. *Frozen teacher*
>
> We agree with the reviewer that for any distillation pretraining recipes, one of the technical challenges is to build a strong student model that is not bounded in performance by the teacher. To address this, we proposed the bottleneck scheme, where the student reconstructs teacher embeddings from its bottleneck layer. Even when masking and generative objectives are removed, the bottleneck architecture alone achieves a 0.7% EER improvement over a standard WavLM-Large baseline on the same data (Section 6.6). This demonstrates that our architectural choices serve as a superior prior for the deepfake data manifold when compared to standard distillation. Additionally, regarding pretraining targets, we followed a similar design as WavLM and HuBERT, where the pretraining targets are frozen. While we have ablated on using EMA to update the teacher model instead of keeping it frozen, we found this leads to pretraining stability issues.
>
> 4. *Initialization strategy*
>
> For both 400M and 1B versions, we experimented with random initialization and initialization from pre-trained checkpoints. With the 400M version, the pre-trained initialization leads to slightly better performance (0.5% EER difference across the 56 datasets). With the 1B version, the average EER difference is not significant (<0.1% EER).
>
> 5. *Masking and prediction loss*
>
> For each checkpoint from mask-ratio ablation (i.e., 0% - 50%), we ran evaluations on downstream task with the exact same fine-tuning setup and found 10% to yield the best downstream performance (Appendix Table 11). Additionally, we would like to clarify that even with 0% masking, because of the bottleneck design, the student yielded better performance than the teacher (Section 6.6 - Bottleneck Architecture). Together, these results demonstrate that the student model does not learn an identity mapping.
>
> While the 10% mask ratio is smaller than the conventional practice, because our pretraining data is primarily in-the-wild speech, the reconstruction task is challenging when compared to reconstructing 50% of clean speech, because the model needs to be robust to a variety of real-world noises and learn to model the deepfake nuances. We provide a more detailed discussion in Appendix C.1.
>
> 6. *Loss Balancing*
>
> The relative weights assigned to each loss term were empirically determined using the following ablation studies:
> - A. Only predictive loss
> - B. Only generative loss
> - C. $\lambda=0.1$
> - D. $\lambda=0.25$ (current default)
> - E. $\lambda=0.5$
> - F. $\lambda=0.75$
>
> With A and B, we observed non-convergence with the excluded branch. Among different $\lambda$ trials, we found that when $\lambda<0.25$ the generation loss is decreasing much slower than $\lambda>0.25$, suggesting the dominance of embedding reconstruction loss. Among D-F, the generation loss curves are very similar but larger $\lambda$ values lead to higher embedding reconstruction loss. We will include these details in the Appendix.

---

> > ### Author Rebuttal · Reviewer_2YEr · 2026-04-03
> >
> > I thank the authors for their rebuttal which answered my questions. I think their conceptual contribution is strong: replacing discrete token quantization with continuous multi-layer embedding prediction, coupled with a flow-matching generative branche is a genuinely novel recipe for deepfake detection. I will increase my score to 'Accept'.
> >
> > However, I want to emphasize that the lack of a public release for the code and model weights limits the paper's overall impact. Withholding these assets fundamentally contradicts the purpose of introducing a 'foundational' model to the community.

---

### Official Review · Reviewer_A4Sm · 2026-03-10

**Soundness:** 3
**Presentation:** 3
**Significance:** 3
**Originality:** 3
**Overall Recommendation:** 4
**Confidence:** 3

**Summary:**

To address the issue of insufficient generalization in existing foundation models for speech deepfake detection, this paper proposes a novel speech foundation encoder named Alethia. The authors point out that current self-supervised speech foundation models mainly rely on masked token prediction with discrete targets, and such a quantization process tends to filter out subtle, continuous generative artifacts that are critical for authenticity discrimination. The authors aim to explore the concept of combining continuous embedding prediction with generative pre-training to overcome these fundamental limitations. To achieve this goal, this paper introduces a novel pre-training paradigm consisting of two parallel branches: one is  masked embedding prediction, which reconstructs multi-layer continuous features from a frozen teacher model using a specialized structure to preserve rich fine-grained information; the other is a flow-matching-based spectrogram generation branch, designed to capture subtle anomalies at the low-level acoustic level. In addition, the authors construct a high-quality dataset with 19,000 hours to scale up training and investigate this key concept. Extensive benchmarking is conducted on 5 downstream tasks covering 56 datasets. The results show that Alethia significantly outperforms existing speech foundation models under various complex perturbations and zero-shot cross-domain generalization scenarios.

**Compliance With Llm Reviewing Policy:**

Affirmed.

**Final Justification:**

Thanks for authors, I have no problems.

**Key Questions For Authors:**

1.The pre-training method relies on a pre-trained teacher model. This raises a potential concern, could the performance upper bound of the student model be limited by the inherent feature space of the teacher model? If the general teacher model itself is insensitive to certain specific acoustic artifacts, will the student model inevitably inherit these blind spots when learning its output features?
2.In the classic masked pre-training paradigm, the loss is typically computed only on masked positions to prevent the model from "cheating" by learning trivial identity mappings or information shortcuts. However, this paper finds that computing the loss over all time steps actually improves convergence and reduces prediction error on masked positions.
3.How exactly do gradients on unmasked frames help the student model better infer the content of masked frames? Does this imply that the projection head Φ of the student model is highly unstable in the early training stage and must be forcibly aligned using supervision signals from unmasked frames?

**Limitations:**

yes

**Strengths And Weaknesses:**

Strengths:
1.It identifies the inherent limitations of traditional discretization objectives in capturing continuous deepfake artifacts. By abandoning K-means clustering features and adopting dual supervision via continuous multi-layer embedding prediction plus auxiliary flow-matching generation, this design well meets the dual requirements of deepfake detection for both high-level semantics and low-level acoustic features.
2.The paper provides an extensive evaluation on 56 datasets across 5 related tasks. Among papers in the field of foundation models or detectors, covering such a wide range of cross-domain testing greatly improves the confidence of the conclusions.
3.Against various acoustic perturbations and on unseen generation methods, Alethia achieves performance superior to speech foundation models of similar parameter size.
4.It collected 30,000 hours of diverse speech and introduced a four-stage quality control pipeline including VAD, speaker separation, and MOS evaluation to select 19,000 hours of high-quality data.
5.It possesses zero-shot generalization ability without requiring retraining for specific tasks or new scenarios, and is robust to real-world perturbations, adapting to practical detection requirements in digital communications.

Weakness:
1.The discussion on hyperparameter sensitivity and weight tuning is limited. The final pre-training loss function combines L1 distance, cosine similarity, and flow-matching generation loss. The magnitudes of different losses may vary significantly during training, but the paper does not sufficiently present ablation or sensitivity analyses of the model regarding these critical weight assignments, nor does it discuss dynamic balancing strategies.

---

> ### Author Rebuttal · Authors · 2026-03-31
>
> Thank you for your valuable feedback and positive assessment of our paper. Please find our responses below.
>
> > 1. *"The discussion on hyperparameter sensitivity and weight tuning is limited. . . . dynamic balancing strategies."*
>
> We agree with the reviewer that ablations are crucial for understanding how each component in the pretraining loss affects the overall performance. Among various configurations, the following factors affected the pretraining stability and downstream performance the most (ranked in order). Details on the ablations are presented in the Appendix.
> - Masking strategy and ratios (Appendix C.1)
> - Predictive loss calculation (Appendix C.2)
> - Spectrogram reconstruction method (Appendix C.3)
>
> We agree with the reviewer that further details on loss ablations will strengthen the manuscript. We will add sections covering the following aspects:
> - **Predictive loss and generation loss balancing** ($\alpha$ and $\beta$ in equation 4): Due to the character limit, please see our response to **reviewer 2YEr - point 6**.
> - **L1 and cosine distance ablations**: We performed ablations on the setup including (a) L1 distance only and (b) cosine distance only. We further finetuned these checkpoints and evaluated them downstream over 56 datasets. Regarding pretraining, we have observed that equal balance (1:1) leads to slightly better downstream EER (~0.5% EER difference) compared to (a) and (b). We will incorporate detailed numbers in the Appendix of the revised manuscript.
>
> > 2. *"The pre-training method relies on a pre-trained teacher model. . . . learning its output features?"*
>
> In distillation-guided pretraining, one of the key challenges is to learn a strong student model that can outperform the teacher. We studied different distillation methods and discussed the limitation of how existing methods could lead to performance bounded by the teacher model (Section 4.1.3 - Bottleneck architecture). Due to the character limit, we encourage the reviewer to refer to our responses to **reviewer 2YEr (point 3)** and **reviewer cnqW (point 4)**, for a more thorough discussion on why and how the proposed bottleneck distillation method tackles this limitation. Regarding the point on student model inheriting the teacher model’s blind spots, we acknowledge that this could potentially be an issue for all distillation-guided pretraining methods. One way to tackle this is to perform several epochs of pretraining, where later epochs use earlier epoch checkpoints as the new teachers to iteratively reduce blind spots. We thank the reviewer for raising this point. We will include the above notes in the limitation and future work section.
>
>
> > 3. *“In the classic masked pre-training paradigm . . . the content of masked frames? Does this imply that . . . signals from unmasked frames?”*
>
> We thank the reviewer for the careful read and insightful observation. We initially explored the default masked prediction scheme and found via experimentation that **the pretraining stability is dependent on two factors: masking ratio and where the predictive loss is calculated**. These two factors need to be adjusted according to how noisy the pretraining data is. In our case, the pre-training data is dominated by in-the-wild speech where data perturbations are much more severe than the relatively cleaner  pre-training data used for standard SFMs such as Wav2vec/WavLM/HuBERT. As a result, we noticed that a masking ratio of  50% and the loss calculation on masked-only positions leads to much slower convergence when compared to using clean speech for pretraining. We reduced the masking ratio down to 10% and adjusted the loss calculation correspondingly to stabilize pretraining. More details can be found in Appendix C.1 and C.2.

---

> > ### Author Rebuttal · Reviewer_A4Sm · 2026-04-01
> >
> > Thanks for the authors. I will keep my score.

---

### Official Review · Reviewer_qjgQ · 2026-03-12

**Soundness:** 2
**Presentation:** 2
**Significance:** 2
**Originality:** 2
**Overall Recommendation:** 4
**Confidence:** 3

**Summary:**

This paper proposes an audio encoder training method specialized for detecting generated audio. The method uses an flow matching based decoder for the reconstruction loss, and also utilizes a student-teacher loss for intermediate representations. The input audio to the encoder is masked.

**Compliance With Llm Reviewing Policy:**

Affirmed.

**Final Justification:**

Rebuttal is adequate and paper has its merits as I indicated in my review.

**Key Questions For Authors:**

- What makes the proposed architecture especially well suited for detecting deep fakes?

- Did you try your encoder on other downstream tasks other than deepfake detection? It might be interesting to document this.

- In table 1, under expanded + aug setting Alethia Base seems to be performing worse than Wav2vec2-2b. How many parameters do you have in Alethia-base and Alethia-large?
Also, do you have a sense how do the performance distribution change across different datasets? As far as I understand the results in table 1 are on an aggregate dataset, so it would be good to report the performance distribution.

**Limitations:**

Yes

**Strengths And Weaknesses:**

The paper attacks an important problem. The proposed methodology seems to be novel, though not ground breaking.

One aspect that I am struggling with what makes the proposed architecture specifically well-suited for generated audio.

The paper is generally well written enough to follow, but could be improved in some parts.

The experiments seem to be quite comprehensive, and we observe a performance improvement across a variety of datasets pertaining to generated audio detection.

---

> ### Author Rebuttal · Authors · 2026-03-30
>
> Thank you for your valuable feedback and positive assessment of our paper. Please find our responses below.
>
> > 1. *"What makes the proposed architecture especially well suited for detecting deep fakes?"*
>
> We made two important design choices to tailor the proposed pretraining algorithm for deepfake detection. Firstly, we propose to predict continuous embeddings instead of discretized targets to avoid information loss (Section 4.1.3). This choice is motivated in Section 3, where we demonstrate the limitation of existing pretraining methods, specifically how using discrete targets leads to information loss, and that deepfake traces are potentially overlooked during the discretization process. Secondly, we include a generation branch that injects low-level acoustic attributes into the learned representations, in order to complement the information learned via the predictive branch. The generation branch reconstructs spectrograms from the learned representations, and allows the pretrained model to capture acoustic attributes necessary to identify deepfakes.
>
> > 2. *"Encoder performance on other downstream tasks"*
>
> We agree with the reviewer that the proposed pretraining method could generalize well to other tasks, particularly those that rely on in-the-wild speech. While we are limited by the time of rebuttal phase, we plan to evaluate performance on the SUPERB benchmark, which constitutes a wide variety of speech tasks.
>
> > 3. *"In table 1, under expanded + aug setting Alethia Base seems to be performing worse than Wav2vec2-2b. How many parameters do you have in Alethia-base and Alethia-large?"*
>
> Alethia-Base has 400M and Alethia-Large has 1B parameters (Section 4.1.1). We present results for the two versions  to facilitate a fair comparison between models of similar size.  We note from Table 4 that Alethia-Base (400M) outperforms Wav2vec-300M, HuBERT-Large, and WavLM-Lage, while Alethia-Large (1B) outperforms Wav2vec-1b.
>
> > 4. *"Also, do you have a sense how do the performance distribution change across different datasets? As far as I understand the results in table 1 are on an aggregate dataset, so it would be good to report the performance distribution."*
>
> We agree that a dataset-level performance breakdown will facilitate a deeper understanding.  We have tabulated the per-dataset results in Tables 16 and 17, and shown the  relative improvement over baselines in Figures 5 and 6 (all in the Appendix).

---

> > ### Author Rebuttal · Reviewer_qjgQ · 2026-04-02
> >
> > The rebuttal that the authors provide is adequate. I will keep my score.

---

### Decision · Program_Chairs · 2026-04-30

**Decision:**

Accept (regular)

**Comment:**

This paper introduces Alethia, a foundational audio encoder specifically tailored for voice deepfake detection and localization. It departs from standard speech foundation models (SFMs) by replacing discrete masked token predictions with continuous bottleneck masked embedding prediction, combined with a flow-matching-based spectrogram reconstruction branch. The model is trained on a large-scale 19k-hour curated dataset and extensively evaluated across 5 tasks and 56 datasets.

The rebuttal successfully resolved the primary technical concerns of the reviewers. The paper attacks a highly relevant and pressing problem with a technically sound, dual-branch pretraining recipe. The sheer rigor of the empirical evaluation gives high confidence in the model's efficacy. The contribution is solid and will be of high interest to the ICML community working on audio, foundation models, and AI safety/deepfake detection.